# RAZORATTENTION: EFFICIENT KV CACHE COMPRESSION THROUGH RETRIEVAL HEADS

Hanlin Tang[*][1], Yang Lin[1], Jing Lin[1], Qingsen Han[1], Danning Ke[1], Shikuan Hong[1], Yiwu Yao[1], and Gongyi Wang[1]

[1]Huawei Technologies Co., Ltd

## ABSTRACT

The memory and computational demands of Key-Value (KV) cache present significant challenges for deploying long-context language models. Previous approaches attempt to mitigate this issue by selectively dropping tokens, which irreversibly erases critical information that might be needed for future queries. In this paper, we propose a novel compression technique for KV cache that preserves all token information. Our investigation reveals that: i) Most attention heads primarily focus on the local context; ii) Only a few heads, denoted as retrieval heads, can essentially pay attention to all input tokens. These key observations motivate us to use separate caching strategy for attention heads. Therefore, we propose RazorAttention, a training-free KV cache compression algorithm, which maintains a full cache for these crucial retrieval heads and discards the remote tokens in non-retrieval heads. Furthermore, we introduce a novel mechanism involving a "compensation token" to further recover the information in the dropped tokens. Extensive evaluations across a diverse set of large language models (LLMs) demonstrate that RazorAttention achieves a reduction in KV cache size by over 70% without noticeable impacts on performance. Additionally, RazorAttention is compatible with FlashAttention, rendering it an efficient and plug-and-play solution that enhances LLM inference efficiency without overhead or retraining of the original model.

## 1 INTRODUCTION

Long-context large language models (LLMs) have significantly advanced capabilities in natural language processing across diverse tasks. However, the growth of the Key-Value (KV) cache under increasing input length has become the major bottleneck for deployment. There are been plenty of previous work designed to alleviate this problem by compressing the KV cache size, including quantization (Sheng et al., 2023; Zhao et al., 2024; Lin et al., 2024), token-dropping (Zhang et al., 2023; Xiao et al., 2024), local attention (Jiang et al., 2023; Child et al., 2019), etc.

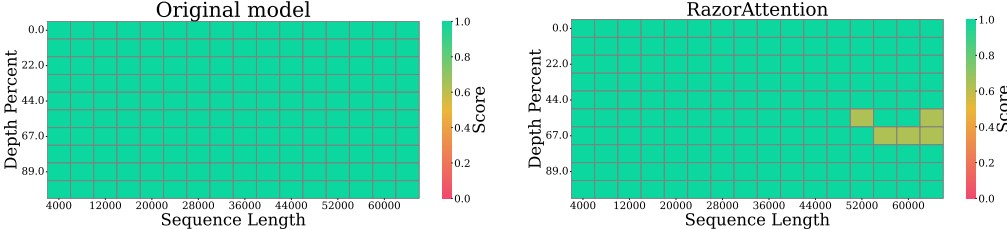

Figure 1: RazorAttention achieves comparable performance to the original model, even with 70% KV cache compressed. To demonstrate this, we tested Llama2-13B-64K (Fu et al., 2024) on the Needle in A Haystack benchmark (gkamradt, 2023).

---

[*]Corresponding author: tanghl1994@gmail.com

> **Input context**: "DOD's MILCON appropriations are used to fund the acquisition, construction, installation...
> Mary's favorite number is 34251... Bob's favorite number is 7690... reviewing project cost estimates."
>
> **Q1: "What is Mary's favourite number?"**
> Original model: "Mary's favorite number is 34251." ✔
> H2O: "Mary's favorite number is not explicitly mentioned in the text provided." ✘
> SnapKV: "Mary's favorite number is 34251." ✔
> RazorAttention: "Mary's favorite number is 34251." ✔
>
> **Q2: "What is Bob's favourite number?"**
> Original model: "Bob's favorite number is 7690." ✔
> H2O: "!!!!!!!!!!!!!!!!!!!!!!!!!!!!!!!!!!!!!!!!!" ✘
> SnapKV: "!!!!!!!!!!!!!!!!!!!!!!!!!!!!!!!!!!!!!!!!!!" ✘
> RazorAttention: "Bob's favorite number is 7690." ✔

Figure 2: Importance-based token-dropping methods cannot work when querying the less relevant information to the main theme. Here, we use an 8K document from LongBench (Bai et al., 2023b) and add two sentences that are not relevant to the main theme. In this case, H2O discards tokens that are less relevant to the main theme, leading to failures in both Q1 and Q2. SnapKV discards tokens based on the first query, making it effective for Q1 but failing in subsequent queries like Q2. Only RazorAttention successfully outputs the exact information from the lengthy input even when we compress 70% of the KV cache.

One major direction for KV cache compression is to directly drop tokens deemed unimportant so far (Zhang et al., 2023; Xiao et al., 2024; Liu et al., 2023b; Li et al., 2024). These methods inherently assume that tokens considered unimportant will not be needed in future queries, which does not hold in practical scenarios. For instance, a user might request information that is not directly aligned with the main theme of the processed text, or engage in a multi-round conversation querying different segments from the context. In these cases, the importance-based token-dropping methods can lead to significant performance degradation since the actual information required by the query might be discarded if considered unimportant (see our example on Qwen1.5-7B-Chat (Bai et al., 2023a) in Figure 2). This leads us to pose a critical question:

*"Can we find a way to reduce the KV cache size without losing semantic information?"*

In this work we address this problem from a novel perspective. Our investigation reveals that there exists a "retrieve and process" mechanism in LLMs when processing a long context. More specifically, LLMs can accurately recall the queried information from a lengthy input through certain group of attention heads, which we denote as "retrieval heads" (see Section 3.3 for definition). These heads are capable of concentrating most of their attention weights on the relevant information (w.r.t. the queries) and increasing the output probability for those words. Another important finding is that non-retrieval heads primarily focus on local context or the attention sink (Xiao et al., 2024), which means these heads cannot effectively utilize all the semantic information from the input. Based on these important findings, we hypothesize that LLM runs the reasoning procedure on a "retrieve and process" basis. That says, the model first uses the retrieval heads to gather relevant information, and then non-retrieval heads to process the retrieved information and generate the final response. This motivates us to design separate caching strategies for different heads: For retrieval heads, we keep the KV cache unaltered; for the rest heads, we only cache recent tokens and attention sinks.

Beyond this, we notice that there still exists a certain accuracy gap when we directly discard all the remote tokens in the non-retrieval heads. Therefore for these non-retrieval heads, we designed a "compensation token" for compressing the dropped cache into one token, and proved that the accuracy degradation due to the truncated KV cache gets further improved with this compensation token. With retrieval heads and compensation tokens, we prove that our algorithm, namely RazorAttention, can successfully compress 70% of the KV cache without noticeable performance degradation as illustrated in Figure 1. RazorAttention can even support the compression of 1024K sequences, achieving nearly lossless precision in the Needle in a Haystack task after compressing the KV cache by 50%, as shown in Figure 5.

Last but not least, previous importance-based token-dropping methods cannot be combined with FlashAttention due to their reliance on the attention weights to compute the importance score, making them impracticable for implementation since FlashAttention is one of the most important components in long-context inference. RazorAttention addresses this problem since it does not use the attention map as the metric. The head-wise pruning criterion is totally compatible with FlashAttention, and the computation overhead of the compensation token is negligible. Therefore RazorAttention could achieve a substantial inference speedup when compared to previous methods.

To the best of our knowledge, RazorAttention is the first training-free token reduction algorithm that achieves a nearly lossless 3X KV cache reduction. We evaluated RazorAttention on models including Qwen (Bai et al., 2023a), Llama-2 (Touvron et al., 2023), Llama-3 (AI@Meta, 2024) and Baichuan (Baichuan, 2023) on long-context tasks to prove its effectiveness. Our contribution can be summarized as follows:

- We systematically analyze the attention dynamic of Transformers under lengthy inputs. Our work reveals that only a few retrieval heads can essentially recall information from the whole input while the rest heads mainly focus on the local context.
- We introduce a novel algorithm, namely RazorAttention, that is capable of reducing the KV cache size by 70% under minimal impact on performance for contexts ranging from 8K to 100K tokens. We designed an accurate and data-free metric for allocating all the retrieval heads, together with an error compensation strategy for compensating the information loss due to the truncated KV cache.
- RazorAttention introduces negligible overhead in compression and is compatible with FlashAttention, rendering it an efficient and plug-and-play solution that enhances LLM inference efficiency without training or significant overhead. Extensive experiments demonstrate that RazorAttention can be effectively applied to various models and tasks.

## 2 RELATED WORK

As the sequence length increases, the memory consumption of KV cache rapidly expands, potentially surpassing the size of the model parameters themselves. This leads to an urgent need for KV cache compression, particularly in scenarios with limited GPU memory. One direction is non-Transformer architecture design, such as Mamba (Gu & Dao, 2024), Mamba2 (Dao & Gu, 2024), Infini-Transformer (Munkhdalai et al., 2024), RWKV (Peng et al., 2023)and Griffin (De et al., 2024). However, in this paper we focus on KV cache reduction for typical Transformers, which is the most widely used model structure. Below we introduce several approaches for KV cache compression.

**Quantization** Quantization is a classic yet effective approach to neural network compression. In the field of LLM Quantization, while the outlier challenge attracts great attention (Xiao et al., 2023; Wei et al., 2022; 2023) to tackle, the application of which on KV cache is often seen as a by-product of activation quantization. Nevertheless, there are several noteworthy works demonstrating the value of KV cache quantization. FlexGen, Atom and QServe (Sheng et al., 2023; Zhao et al., 2024; Lin et al., 2024) carefully designed quantization pipelines that utilize KV cache compression to boost the overall inference throughput. KVQuant (Hooper et al., 2024) integrates several techniques to minimize KV quantization error and KIVI (Zirui Liu et al., 2023) pushed the limit towards 2-bits. Besides the post-training methods, LLM-QAT (Liu et al., 2023a) offers a data-free distillation process that further recovers the performance of the model.

**Token-dropping** Token-dropping methods assume that not all key-value pairs are essential in self-attention computations, so memory usage can be saved by identifying and removing unimportant KV cache. StreamingLLM (Xiao et al., 2024) utilizes sliding window technology, preserving only the KV pairs of attention sink tokens and those within the sliding window, thereby reducing memory footprint and stabilizing model performance. H2O (Zhang et al., 2023) is one of the pioneers that use the attention scores to evaluate the importance of each token, followed by an eviction strategy that greedily selects cache with higher scores. Scissorhands (Liu et al., 2023b) and one of the latest work SnapKV (Li et al., 2024) use similar ideas by narrowing the computation range to consider attention scores related to recent information. Built on that, PyramidKV and PyramidInfer (Cai. et al., 2024; Yang et al., 2024) analyze the attention concentration patterns and further reduce KV

cache in later layers. Moreover, research efforts have been made to understand KV cache from different perspectives: FastGen (Ge et al., 2024) paid attention to special tokens and punctuation, SubGen (Zandieh et al., 2024) investigated the clusterability of key embedding and CORM (Dai et al., 2024) discovered strong correlation amongst tokens of near neighbors.

**Non-MHA Attention**  Another category focuses on reducing KV cache by sharing cache across attention heads. MQA (Shazeer, 2019) aggressively uses a single KV head for all heads, whereas GQA (Ainslie et al., 2023) suggests an intermediate number of heads to balance the trade-off between inference speed and output quality. Furthermore, MLA (DeepSeek-AI et al., 2024) presents a novel caching method by low-ranking KV cache of all heads into single latent space.

Our algorithm is motivated by the idea from Olsson et al. (2022), where the authors noticed that there are certain groups of attention heads, denoted as the induction heads, that can effectively recall the queried information from the input. Recent study (Wu et al., 2024) also validated this property under extended inputs. This is the first work that proposes a head-wise pruning criterion for KV cache compression based on the interpretability of the attention mechanism.

## 3  METHODOLOGY

In this section, we introduce the key components of RazorAttention. We firstly apply RazorAttention to models using ALiBi (Press et al., 2022) positional embedding (denoted as ALiBi models) to provide an intuitive understanding of the retrieval and non-retrieval heads. Afterwards, we demonstrate that models using RoPE (Su et al., 2023) positional embedding (denoted as RoPE models) also exhibit this crucial characteristic, which reveal that KV cache within RoPE models can also be efficiently compressed under minimal loss of accuracy.

### 3.1  RAZORATTENTION FOR ALiBi MODELS

For ALiBi models, its $h$-th attention head computes the attention score according to

$$S_{m \to n}(\boldsymbol{q}; \boldsymbol{k}) = \boldsymbol{q}_m \boldsymbol{k}_n^\intercal - l_h(m - n), \tag{1}$$

where $\boldsymbol{q}_m$ is the query tensor at the $m$-th position, $\boldsymbol{k}_n$ is the key tensor at the $n$-th position, $l_h$ is the head-specific slope, $S_{m \to n}(\boldsymbol{q}; \boldsymbol{k})$ is the attention score. Notice that $(m \geq n)$ is guaranteed by the causality of attention.

In the scenario where $l_h(m - n)$ significantly dominates $\boldsymbol{q}_m \boldsymbol{k}_n^\intercal$, the attention between $\boldsymbol{q}_m$ and $\boldsymbol{k}_n$ would decay to zero, meaning that the contribution of any tokens positioned further than $n$ becomes negligible for the output at position $m$. The following theorem formalizes this observation.

**Theorem 1** *Given an attention head that calculates the attention score as per equation 1, for any $\epsilon \in (0, 1)$, the attention weight from $\boldsymbol{q}_m$ to $\boldsymbol{k}_n$ can be upper bounded by:*

$$Attn_{m \to n}(\boldsymbol{q}; \boldsymbol{k}) = \frac{\exp(S_{m \to n}(\boldsymbol{q}; \boldsymbol{k}))}{\sum_{n=0}^{m} \exp(S_{m \to n}(\boldsymbol{q}; \boldsymbol{k}))} \leq \epsilon, \quad \forall n < m - C_0,$$

$$L_h := \frac{2\|W_{Q_h} W_{K_h}\|_2 \left(\|\boldsymbol{\gamma}\|^2 + \|\boldsymbol{b}\|^2\right) - \log(\epsilon)}{l_h}. \tag{2}$$

*Here $W_{Q_h}$ and $W_{K_h}$ are the query and key matrices of the $h$-th attention head, $\gamma$ and $\boldsymbol{b}$ are the weight and bias for the LayerNorm layer before attention ($\boldsymbol{b} = \boldsymbol{0}$ for RMSNorm (Zhang & Sennrich, 2019)), and $\|\cdot\|_2$ denotes the $l_2$-norm of the matrix. $L_h$ can be viewed as the vision scope of the head. The detailed proof can be found in Appendix A.1.*

Theorem equation 1 indicates that when the distance between $\boldsymbol{q}_m$ and $\boldsymbol{k}_n$ exceeds $C_0$, the attention weight between these two tokens falls below $\epsilon$. When $\epsilon$ is sufficiently small (e.g., 0.1%), remote tokens impose minimal influence on the final output and can thus be discarded. Building on this principle, ALiBi models dynamically adjust the KV cache size for each head. We first compute the effective attention scope $L_h$, and keep only the recent $L_h$ tokens in the KV cache, since any token further than $L_h$ impose attention weight no more than $\epsilon$, we can safely discard them for compression. Therefore, for ALiBi models, the retrieval heads are the ones with a larger $L_h$, while the non-retrieval heads has a smaller attention vision $L_h$.

## 3.2 RAZORATTENTION FOR ROPE MODELS

For RoPE models, each attention head computes the attention score according to

$$S_{m \to n}(\boldsymbol{q}; \boldsymbol{k}) = \boldsymbol{q}_m \boldsymbol{k}_n^{\mathsf{T}}, \quad \boldsymbol{q}_m = \mathcal{R}_m \boldsymbol{q}, \quad \boldsymbol{k}_n = \mathcal{R}_n \boldsymbol{k} \tag{3}$$

Where $q_m$ and $k_n$ are the query and key states after the rotary transformation, and $R_m$ and $R_n$ are the rotary matrices at positions $m$ and $n$ (see Su et al. (2023) for details). Although RoPE embedding does not inherently suggest long-range decaying attention, our empirical findings indicate that only about 15% of the heads, which we term "retrieval heads," are capable of effectively utilizing long-range information to ensure the model's reasoning performance, while the remaining heads primarily focus on local context. As shown in Table 1, when all heads are set to focus only on local context (Protect 0% heads), the accuracy significantly decreases by an average of 7.3%; when 15% of the heads are randomly selected to attend to long-range information (Protect 15% heads Random) while the rest focus on local context, the accuracy significantly decreases by an average of 7.1%; however, when retrieval heads are selected to focus on long-range information (Protect 15% heads Retrieval) while the remaining heads focus on local context, the accuracy is comparable to the baseline, with only a decline of 0.3%.

| Protection heads | MultiFieldQA-en | Hotpotaqa | 2Wikimqa |
|---|---|---|---|
| Baseline | 46.94% | 50.96% | 36.36% |
| Protect 0% heads | 37.36% | 42.36% | 31.33% |
| Protect 15% heads (Random) | 38.33% | 42.46% | 31.97% |
| Protect 15% heads (Retrieval) | 46.66% | 50.49% | 36.12% |

Table 1: We protected the KV cache within different groups of attention heads while keeping only the recent 4K tokens in the rest with Qwen1.5-7B. The results indicate that both unprotected heads and randomly protected heads yield similar poor performance, while only protecting the retrieval heads effectively retains most of the LLM's performance. This clearly shows that most attention heads rely solely on local context, and only retrieval heads can fully utilize all contextual information.

Based on the findings above, we directly decrease the KV cache for all non-retrieval heads. The performance of the model is mostly retained as shown in Table 1.However, a notable accuracy gap remains, indicating that some information is still being lost. Moreover, the test result on Needle in a Haystack shows a clear performance degradation even when we protect the KV cache of retrieval heads (see our ablation result in Figure 8). To further improve performance, we designed a lightweight and effective way to compress the information in the dropped token into a "compensation token". The compensation token is defined as

$$\hat{\boldsymbol{k}} = \frac{1}{N_d} \sum_{m \in \{\mathcal{D}\}} \boldsymbol{k}_m, \quad \hat{\boldsymbol{v}} = \frac{1}{N_d} \sum_{m \in \{\mathcal{D}\}} \boldsymbol{v}_m. \tag{4}$$

Here $\hat{\boldsymbol{k}}$, $\hat{\boldsymbol{v}}$ are the compensation tokens for the dropped KV cache, $\{\mathcal{D}\}$ contains the indices of the dropped tokens and $N_d$ is the number of the dropped tokens. Afterward, we discard the dropped tokens and augment the KV cache with the compensation token $\hat{\boldsymbol{k}}$ and $\hat{\boldsymbol{v}}$, where $\{K, V\}$ are the KV cache of the remaining token after rotary transformation. Denoting the compressed KV cache as $\{K, \hat{\boldsymbol{k}}\}$ and $\{V, \hat{\boldsymbol{v}}\}$, the attention output of the current token follows

$$\text{Attn}(\boldsymbol{q}_m, \{K, \hat{\boldsymbol{k}}\}, \{V, \hat{\boldsymbol{v}}\}) = \frac{N_d \exp\left(\boldsymbol{q}_m \hat{\boldsymbol{k}}^{\mathsf{T}}\right) \hat{\boldsymbol{v}} + \sum_{n \notin \{\mathcal{D}\}} \exp\left(\boldsymbol{q}_m \boldsymbol{k}_n^{\mathsf{T}}\right) \boldsymbol{v}_n}{N_d \exp\left(\boldsymbol{q}_m \hat{\boldsymbol{k}}^{\mathsf{T}}\right) + \sum_{n \notin \{\mathcal{D}\}} \exp\left(\boldsymbol{q}_m \boldsymbol{k}_n^{\mathsf{T}}\right)}. \tag{5}$$

In Figure 3(a) we provide an illustrative example of RazorAttention for RoPE models. With compensation tokens, the accuracy is further improved, making RazorAttention almost lossless even dropping 70% of the KV cache in the non-retrieval heads. Below we introduce how we determine the retrieval heads group.

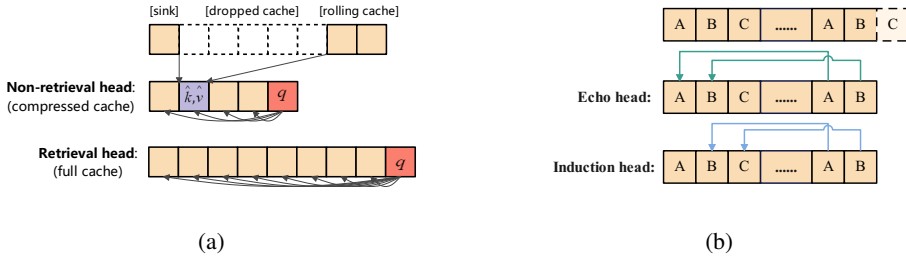

(a)                                                (b)

Figure 3: In Figure 3(a) we present the illustration of how RazorAttention compress the KV cache. For retrieval heads, we maintain a full cache for retaining all the tokens' information. For non-retrieval heads, we directly discard remote tokens and compress the discarded tokens into a compensation token whose KV cache is denoted as $\{\hat{\boldsymbol{k}}, \hat{\boldsymbol{v}}\}$. In Figure 3(b) we provide an illustration example of the echo head and induction head. The current token is "B" and the generated token is "C". In this case, the echo head would mainly attend to token "B" while the induction head mainly attend to token "C" in previous context.

| Hyper-parameter | Settings |
|---|---|
| Buffer length | $\max(4000, N/5)$ |
| Induction head protection | top $14\%$ |
| Echo head protection | top $1\%$ |
| Sink token num | 4 |

Table 2: General hyper-parameter settings for experiments in the paper, which leads to 3.125x compression of KV cache under long context input.

## 3.3   IDENTIFICATION OF RETRIEVAL HEADS

For ALiBi models, the attention scope can be directly determined via equation 2 and KV cache can be dropped accordingly. However, for RoPE models, the retrieval heads need to be identified in a more sophisticated way. Our investigation reveals that two groups of heads are essential in processing long context, so both of them should be included as retrieval heads as stated below.

- **Echo head**: The head tends to attends back to previous token (referred as echo token) identical to the current token.

- **Induction head**: The head tends to attend to the previous token (namely induction token) that is immediately succeeded by the current token. Basically it attends to the coming token that also exists in previous context.

In Figure 3(b) we present an illustrative example explaining the echo heads and induction heads. In order to identify the retrieval heads, we generate $K$ (for example, $K = 2500$) random tokens, repeat these tokens 4 times, and then use it as the input of the model. This design minimizes semantic dependencies among tokens, thereby allowing a clearer observation of the behavior of echo and induction heads.

Subsequently, we calculated the echo score (attention weight to the echo token) and induction score (attention weight to the induction token) of all words across all heads. The selection of retrieval heads involves the top-$14\%$ attention heads with the highest induction score and top-$1\%$ of attention heads with the highest echo score (see Table 2). Notice that although we only use much fewer echo heads than retrieval heads, our investigation indicates that both heads are crucial for the retrieving performance for LLMs (see Section 4.2 for ablation results).

With the retrieval heads being identified, we hereby introduce RazorAttention for RoPE Models in Algorithm 1.

---

**Algorithm 1** RazorAttention for RoPE Models

---

**Input:** Non-retrieval headset $\{H\}$, original KV cache (after rotary transformation) $\{K, V\}$, compression ratio $C$, compression threshold $S_0$, sink token num $N_0$.

1: **for** non-retrieval head $h \in \{H\}$ **do**
2:  Compute the buffer length $L_h = \max\left(S_0, \frac{N}{C}\right)$, here $N$ is the number of tokens in the head.
3:  Keeping only the recent $L_h$ tokens near output and first $N_0$ sink tokens, discarding the remaining tokens and compress them into a compensation token according to equation 4.
4: **end for**
5: Non-retrieval heads compute attention according to equation 5, while retrieval heads follow the original attention.

**Output:** Generated output tokens.

---

## 4  EXPERIMENTS

A variety of recent-released LLMs are selected to validate our proposals, including Qwen (Bai et al., 2023a), Llama2 (Touvron et al., 2023), Llama3 (AI@Meta, 2024) and Baichuan (Baichuan, 2023). The selected models are evaluated on Longbench (Bai et al., 2023b) and Needle In A Haystack (gkamradt, 2023) to demonstrate their capabilities in long-context circumstances. The experiments are conducted on NVIDIA GeForce RTX 4090 (24GB). We will first validate the effectiveness of our proposal on various tasks, followed by the ablation study of each component in our algorithm design. Unless explicitly stated, we use RazorAttention with the hyper-parameters as in Table 2. We use H2O (Zhang et al., 2023) and StreamingLLM (Xiao et al., 2024) for comparison. Notice that we do not include SnapKV (Li et al., 2024) as the baseline because it assumes that the query is known before compression, which does not hold in general cases or in a multi-round conversation where the user might query different information from the context (as discussed in Section 1).

### 4.1  ACCURACY EVALUATION

#### 4.1.1  LONGBENCH EVALUATION

In Table 3 we present the results of different algorithms on LongBench (Bai et al., 2023b), which provides a comprehensive assessment to evaluate long-context related abilities of LLMs. We use Qwen1.5-7B and Qwen1.5-72B for testing since they are RoPE models with a context length of 32K. We also include Llama3-8B to validate the performance of RazorAttention on GQA models. The compression settings are shown in Table 2. We choose Baichuan2-13B to demonstrate the effectiveness of RazorAttention on ALiBi models. $\epsilon$ is set to 0.001, the contribution of the current token in the attention map is less than 0.001 under the Alibi encoding. The distance position is calculated according to formula (2). It can be seen that RazorAttention achieved a superior performance across all models compared to StreamingLLM and H2O. The compelling outcomes indicate that RazorAttention can achieve comparable performance as the uncompressed baseline, even under 3X compression ratio.

Moreover, we test Llama3-8B-Instruct as a GQA instance where every 4 attention heads share a single set of KV cache. Hence, we consider the attention heads in a group as all retrieval if one or more heads satisfy inductive or echoing property. The results in Table 3 clearly prove that RazorAttention still work for GQA models.

#### 4.1.2  NEEDLE IN A HAYSTACK EVALUATION

In Figure 4, we present the results on Needle In A Haystack. We use Llama2-7B-80K from Fu et al. (2024) since the context length of this model is 80K. Unlike H2O, whose performance is severely degraded under long inputs, RazorAttention can still accurately recall the queried information. This is a strong evidence proving that RazorAttention can retain all the semantic information within the original context, while importance-based methods inevitably discard information that might be useful in future queries. The compression settings are shown in Table 2.



Figure 4: Performance comparison of RazorAttention and other compression algorithms on Llama2-7b-80K, Needle In A Haystack. Notice that H2O is incompatible with FlashAttention so we get OOM errors when tested on longer sequences, and its performance has already become unusable in this case.

| LLMs | | NrtvQA | Qasper | MF-en | MF-zh | HotpotQA | 2WikiMQA | Musique | GovReport | QMSum | MultiNews | VCSUM | TREC | TriviaQA | LSHT | Lcc | Average |
|---|---|---|---|---|---|---|---|---|---|---|---|---|---|---|---|---|---|
| Qwen1.5-7B-Chat | All KV | 17.58 | 43.16 | 46.94 | 60.98 | 50.96 | 36.36 | 27.86 | 28.78 | 23.24 | 24.02 | 13.91 | 17.64 | 83.77 | 16.96 | 48.35 | 36.03 |
| | StreamingLLM | 6.22 | 24.62 | 18.9 | 34.51 | 20.68 | 12.31 | 5.88 | 3.86 | 3.52 | 20.74 | 3.17 | 8.5 | 36.57 | 13 | 42.51 | 17.00 |
| | H2O | 16.5 | 38.15 | 40.22 | 51.46 | 50.19 | 35.69 | 27.12 | **28.42** | 22.00 | 22.70 | **14.03** | 18.25 | 83.72 | **16.4** | 47.54 | 34.16 |
| | RA | **16.63** | **43.1** | **46.66** | **61.08** | **50.49** | **36.1** | **28.79** | 26.68 | **22.59** | **23.96** | 13.83 | **20.87** | **83.83** | 15.66 | **47.85** | **35.87** |
| Qwen1.5-72B-Chat | All KV | 28.32 | 46.73 | 48.25 | 63.41 | 55.91 | 46.23 | 34.56 | 32.47 | 22.69 | 24.86 | 15.61 | 71.0 | 91.15 | 46 | 65.05 | 46.15 |
| | StreamingLLM | 9.57 | 28.33 | 19.06 | 34.98 | 25.32 | 13.42 | 10.08 | 4.11 | 3.79 | 21.1 | 3.74 | 43.0 | 43.72 | 20.5 | 53.6 | 22.29 |
| | H2O | **27.98** | 41.45 | 43.69 | 55.93 | 54.77 | 45.16 | **34.61** | 32.24 | 22.35 | 24.36 | 14.5 | 70.0 | 91.15 | 42 | 64.2 | 44.29 |
| | RA | 27.97 | **46.44** | **47.36** | **63.04** | **55.92** | **46.15** | 34.36 | **32.35** | **22.75** | **24.91** | **15.17** | **71.0** | **91.49** | 46 | **64.68** | **45.97** |
| Llama3-8B-Instruct | All KV | 21.84 | 37.04 | 45.07 | 52.34 | 44.63 | 27.28 | 23.04 | 28.18 | 24.54 | 26.26 | 14.41 | 0 | 85.90 | 3 | 30.17 | 35.44 |
| | StreamingLLM | 0.61 | 16.29 | 13.41 | 20.05 | 2 | 5.84 | 5.22 | 4.63 | 18.89 | 2.52 | - | 11.54 | - | 26.83 | 9.86 |
| | H2O | 21.14 | 34.1 | 40.84 | 47.13 | 43.47 | **27.13** | 21.31 | 22.85 | 16.36 | 22.3 | 14.52 | - | **86.17** | - | **30.26** | 32.89 |
| | RA | **21.16** | **36.22** | **42.88** | **51.93** | **44.07** | 26.89 | **22.03** | **26.56** | **23.86** | **25.83** | **15.69** | - | 85.83 | - | 30.25 | **34.86** |
| Baichuan2-13B | All KV | 18.63 | 30.16 | 44.1 | 50.36 | 37.93 | 32.62 | 13.90 | 26.16 | 20.14 | 24.58 | 15.66 | 62.5 | 86.61 | 27.5 | 55.36 | 36.41 |
| | StreamingLLM | 5.12 | 12.44 | 23.53 | 32.52 | 16.93 | 16.08 | 6.53 | 5.53 | 1.03 | 5.6 | 3.94 | 42.22 | 30.15 | 7.32 | 35.42 | 16.27 |
| | H2O | 17.81 | 29.89 | **43.74** | 49.54 | **37.02** | 31.71 | 13.54 | 25.8 | 18.96 | 23.31 | 15.11 | 62.41 | 85.25 | 26.86 | 54.45 | 35.69 |
| | RA | **18.22** | **31.87** | 43.6 | **51.36** | 36.97 | **32.89** | **13.98** | 25.51 | **20.13** | **24.51** | 15.41 | 62.5 | **87.23** | 28 | **54.53** | **36.45** |

Table 3: Performance comparison of RazorAttention and other compression algorithms across various LLMs on LongBench. Notice that the performance of Llama3-8B-Instruct on TREC and LSHT are not applicable (close to 0), hence we do not include their result on Llama3-8B.

In Figure 5, we verified that the RazorAttention algorithm can support ultra-long sequences by testing the Needle in a Haystack task using GLM-9B-1M (GLM et al., 2024). It is observed that RazorAttention achieves nearly lossless accuracy at the 1024K sequence length. To ensure effective retrieval of needles in ultra-long sequences, we retained a higher number of retrieval heads (28% induction heads + 2% echo heads).

In Figure 6, we compare the performance of RazorAttention with SnapKV (Li et al., 2024). SnapKV is a competitive approach for KV cache compression, particularly effective when user queries are known in advance. We present SnapKV's performance results in both query-aware and query-agnostic settings. In query-agnostic scenarios, we selected the observation window while disregarding the query tokens, so the user query is not pre-defined, and SnapKV's performance deteriorates significantly. This performance degradation is attributed to SnapKV's reliance on predefined queries. In contrast, RazorAttention demonstrates superior resilience due to its training-free design and headwise sparse pattern, which minimize information loss across a variety of query types.

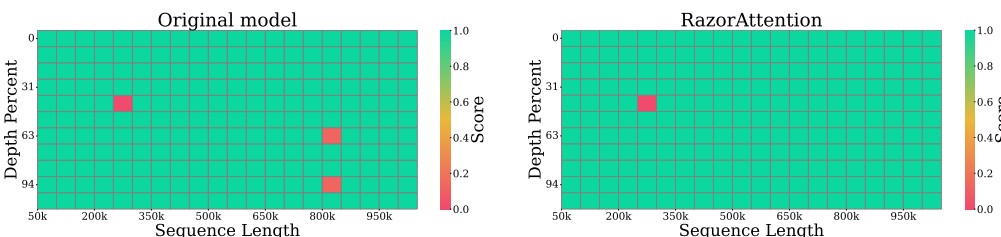

Figure 5: GLM-9B-1M on the 1024K Needle In A Haystack task .



Figure 6: Comparison of RazorAttention and SnapKV on the Llama 3.1-70B model for Needle-in-a-Haystack task accuracy. SnapKV shows significant accuracy differences between query-aware and query-agnostic settings, as shown in Figure 6(a) and Figure 6(a), while RazorAttention consistently maintains higher accuracy, as shown in Figure 6(c).

| Protection scheme | Score |
|---|---|
| 1% Echo + 5% Induction Head | 69.54% |
| 1% Echo + 8% Induction Head | 78.40% |
| 1% Echo + 11% Induction Head | 84.55% |
| 1% Echo + 14% Induction Head | 86.59% |
| Baseline | 87.05% |

Table 4: Qwen1.5-7B-Chat using RazorAttention with different numbers of heads protected, tested on Needle in A Haystack.

## 4.2 ABLATION STUDIES

Below we present the ablation results of RazorAttention, and prove that the algorithm design and configuration are optimally chosen to achieve a higher compression ration with acceptable performance degradation.

### 4.2.1 IMPORTANCE OF ECHO HEADS

Although we only include $1\%$ echo heads in RazorAttention, we notice that this group of heads is quite essential in retrieving information under long context as shown in Figure 7. One possible explanation is that the induction heads depend on the existence of echo heads as discussed in Olsson et al. (2022).

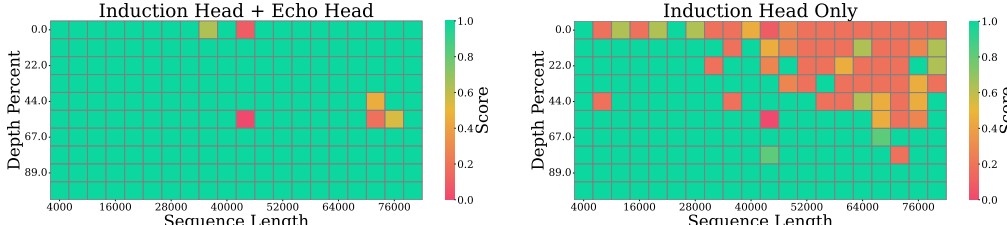

Figure 7: Adding $1\%$ of the echo heads can significantly enhances the retrieving performance of RazorAttention on Llama2-7B-80k.

### 4.2.2 NUMBER OF INDUCTION HEADS

To determine the optimal number of induction heads to use in RazorAttention, in Table 4 we present the accuracy of RazorAttention under various numbers of induction heads. The results show that the accuracy improves continuously with an increasing number of induction heads. We decide to include $14\%$ of the induction heads in order to achieve an optimal balance between the compression ratio and model performance.

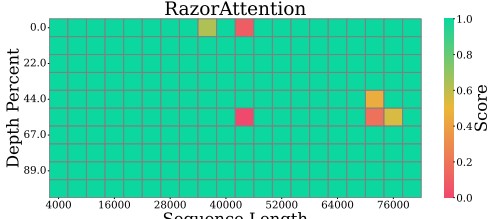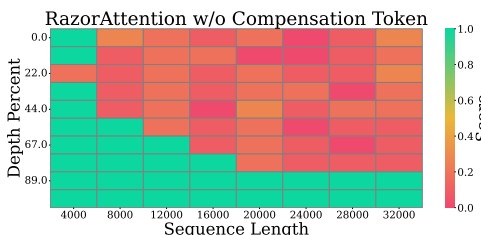

Figure 8: The compensation token is critical for recovering the information loss introduced by the truncated KV cache.

### 4.2.3 IMPORTANCE OF THE COMPENSATION TOKEN

In Figure 8, it is clearly demonstrated that compensation tokens are critical for the performance of RazorAttention. The compensation tokens successfully compressed most of the information from the dropped tokens,thereby maintaining high accuracy even with significant KV cache reduction.

## 5 CONCLUSION

In this paper, we propose RazorAttention, a novel KV cache compression algorithm, which successfully achieves a 3X compression ratio for models use RoPE or ALiBi embeddings. Unlike previous importance-based token-dropping methods which inevitably discard semantic information, RazorAttention preserves all semantic information within retrieval heads. We demonstrate that remote tokens can be effectively compressed into compensation tokens within non-retrieval heads. Furthermore, our head-wise pruning criterion is fully compatible with FlashAttention, making RazorAttention a plug-and-play compression method that accelerates the inference of LLMs under extended context. Our experiments demonstrate that RazorAttention can achieve comparable performance with the original model and surpasses previous methods in both accuracy and efficiency.

## 6 LIMITATION

However, there are still certain limitations of our work. The first question is why attention heads in LLMs behave so differently and how retrieval heads operate under lengthy inputs. The second challenge lies in achieving a higher compression ratio. Although we have successfully reduced the KV cache by 70%, we believe this number can be further improved. Moreover, although we have tested our algorithm on several models, the optimal configuration on other models might be different, meaning that we might need more or less retrieval heads under different cases. These topics are quite important and we will keep investigating them in the future work.

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

# A  APPENDIX

## A.1  PROOF OF THEOREM 1

Below we first give an upper bound for the product of the queries and keys, and then show that the attention weight would decay to zero when the positional bias is significantly larger than that upper bound. Since we have $\boldsymbol{q} = W_{Q_h}\boldsymbol{x}$ and $\boldsymbol{k} = W_{K_h}\boldsymbol{x}$ where $\boldsymbol{x}$ is the input of the Attention block, this leads to

$$\boldsymbol{q}\boldsymbol{k}^\mathsf{T} = \boldsymbol{x}W_{Q_h}W_{K_h}\boldsymbol{x}^\mathsf{T} \leq \|W_{Q_h}W_{K_h}\|_2\|\boldsymbol{x}\|^2. \tag{6}$$

Since $\boldsymbol{x}$ is attained after LayerNorm, which means

$$\boldsymbol{x} = \boldsymbol{\gamma} \odot \frac{\hat{x} - \mu}{\sigma} + \boldsymbol{b},$$

$$\mu = \frac{1}{d}\sum_{i=1}^{d}\hat{x}_i, \quad \sigma = \frac{1}{d}\sum_{i=1}^{d}(\hat{x}_i - \mu)^2.$$

Here $\hat{x}$ is the input of LayerNorm, $d$ is its dimension and $\hat{x}_i$ is the $i$-th dimension of $\hat{x}$. The equation above leads to

$$\begin{aligned}
\|\boldsymbol{x}\|^2 &= \left\|\boldsymbol{\gamma} \odot \frac{\hat{x} - \mu}{\sigma} + \boldsymbol{b}\right\|^2 \\
&\leq 2\left\|\boldsymbol{\gamma} \odot \frac{\hat{x} - \mu}{\sigma}\right\|^2 + 2\|\boldsymbol{b}\|^2 \\
&\leq 2\|\boldsymbol{\gamma}\|^2 + 2\|\boldsymbol{b}\|^2.
\end{aligned} \tag{7}$$

Combining equation 6 and equation 7 we get

$$\boldsymbol{q}\boldsymbol{k}^\mathsf{T} \leq \|W_{Q_h}W_{K_h}\|_2 \left(2\|\boldsymbol{\gamma}\|^2 + 2\|\boldsymbol{b}\|^2\right) \tag{8}$$

In order to give an upper bound for the attention weight, we have

$$\begin{aligned}
\text{Attn}_{m \to n}\left(\boldsymbol{q}; \boldsymbol{k}\right) &= \frac{\exp\left(S_{m \to n}\left(\boldsymbol{q}; \boldsymbol{k}\right)\right)}{\sum_{n=0}^{m} \exp\left(S_{m \to n}\left(\boldsymbol{q}; \boldsymbol{k}\right)\right)} \\
&= \frac{\exp\left(\boldsymbol{q}\boldsymbol{k}^\mathsf{T} - l_h(m - n)\right)}{\sum_{n=0}^{m} \exp\left(S_{m \to n}\left(\boldsymbol{q}; \boldsymbol{k}\right)\right)} \\
&\leq \frac{\exp\left(\boldsymbol{q}\boldsymbol{k}^\mathsf{T} - l_h(m - n)\right)}{\exp\left(S_{n \to n}\left(\boldsymbol{q}; \boldsymbol{k}\right)\right)} \\
&\leq \frac{\exp\left(\boldsymbol{q}\boldsymbol{k}^\mathsf{T} - l_h(m - n)\right)}{\exp\left(\boldsymbol{q}\boldsymbol{q}^\mathsf{T}\right)} \\
&\leq \exp\left(\boldsymbol{q}\boldsymbol{k}^\mathsf{T} - l_h(m - n)\right) \\
&= \frac{\exp\left(\boldsymbol{q}\boldsymbol{k}^\mathsf{T}\right)}{\exp\left(l_h(m - n)\right)}.
\end{aligned}$$

Therefore to ensure $\text{Attn}_{m \to n}\left(\boldsymbol{q}; \boldsymbol{k}\right) \leq \epsilon$, which is equivalent as $\log\left(\text{Attn}_{m \to n}\left(\boldsymbol{q}; \boldsymbol{k}\right)\right) \leq \log(\epsilon)$, we need

$$\log\left(\text{Attn}_{m \to n}\left(\boldsymbol{q}; \boldsymbol{k}\right)\right) \leq \boldsymbol{q}\boldsymbol{k}^\mathsf{T} - l_h(m - n) \leq \log(\epsilon)$$

Taking equation 8 into the equation above, we get

$$\|W_{Q_h}W_{K_h}\|_2 \left(2\|\boldsymbol{\gamma}\|^2 + 2\|\boldsymbol{b}\|^2\right) - l_h(m - n) \leq \log(\epsilon),$$

which gives us

$$m - n \geq \frac{2\|W_{Q_h}W_{K_h}\|_2 \left(\|\boldsymbol{\gamma}\|^2 + \|\boldsymbol{b}\|^2\right) - \log(\epsilon)}{l_h}.$$

In this case, we have

$$\text{Attn}_{m \to n}\left(\boldsymbol{q}; \boldsymbol{k}\right) \leq \epsilon.$$

## A.2 OTHER ACCURACY EVALUATION

We conducted additional evaluations of RazorAttention on the RULER (Hsieh et al., 2024) and InfiniteBench (Zhang et al., 2024) datasets using the LLaMA3.1-8B-Instruct model to demonstrate its effectiveness. In our experimental setup, we retained 30% of retrieval heads(28% induction heads and 2% echo head) and implemented a compression mechanism for non-retrieval heads when sequences exceeded 4k tokens: we preserved an attention window of size 4 and a local window covering 20% of the sequence length, with the remaining tokens being directly compressed.

| Task | Original Model | RA Model |
|------|---------------|----------|
| Ruler-16k | 92.07 | 92.15 |
| Ruler-32k | 84.93 | 85.90 |

Table 5: RULER Benchmark Results

| Task | Original Model | RA Model |
|------|---------------|----------|
| codedebug | 21.57 | 21.57 |
| ensum | 30.71 | 30.58 |
| endia | 19.50 | 19.50 |
| enqa | 29.09 | 29.50 |
| enmc | 63.31 | 63.31 |

Table 6: InfiniteBench Results

## A.3 EFFICIENCY EVALUATION

We evaluate the decoding latency and throughput of RazorAttention on the GLM-9B-1M model using 8 Ascend 910B NPUs, considering different input lengths for both prefill and decoding, as shown in Table 7. Additionally, we achieve a maximum throughput improvement of 64.2% and 71.81% for input lengths of 128k and 256k, respectively, due to the ability to utilize a larger batch size following KV cache compression.

| Input Length | Batch Size | Prefill Speedup | Decoding Speedup |
|--------------|-----------|-----------------|------------------|
| 128k | 1 | 1.2% | 3.1% |
| | 4 | 5.66% | 15.9% |
| | 8 | 8.03% | 26.67% |
| | 10 | 7.05% | 29.47% |
| | 18 | From OOM to feasible inference | |
| 256k | 1 | 6.58% | 9.2% |
| | 4 | 8.76% | 25.47% |
| | 5 | 9.93% | 29.01% |
| | 9 | From OOM to feasible inference | |

Table 7: The performance acceleration of the GLM-9B-1M model on 8 Ascend 910B NPUs.

