# Appendix

## Section 1: Comparison withSnapKV

We present the results of SnapKV in both query-aware and query-agnostic settings:

• Query-aware: SnapKV demonstrates impressive compression ratios and accuracy under known query scenarios.

• Query-agnostic: In scenarios where the user query is not pre-defined, SnapKV's performance deteriorates significantly, as shown by the following results of Needle in a Haystack:

**Figure 1. Query-Aware SnapKV (acc=100, max_length=32k)**

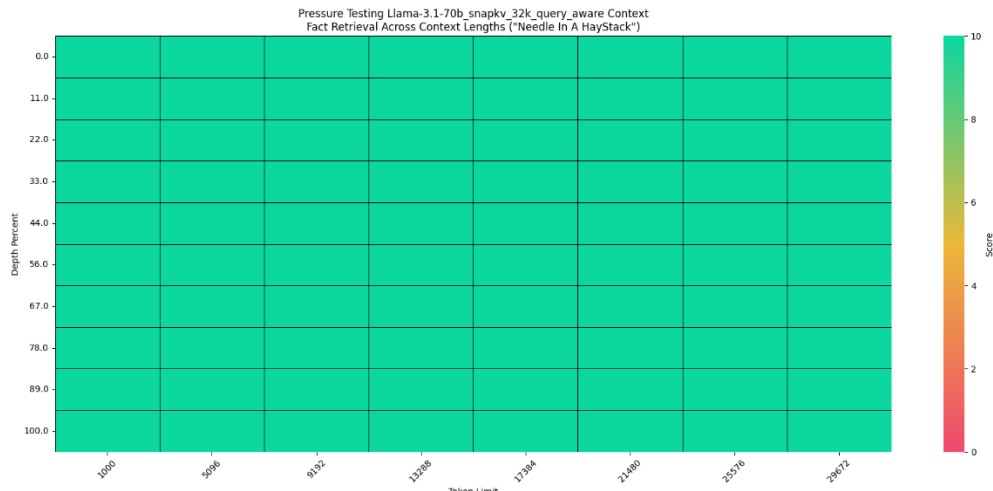

**Figure 2. Query-Agnostic SnapKV(acc=69.75, max_length=32k)**

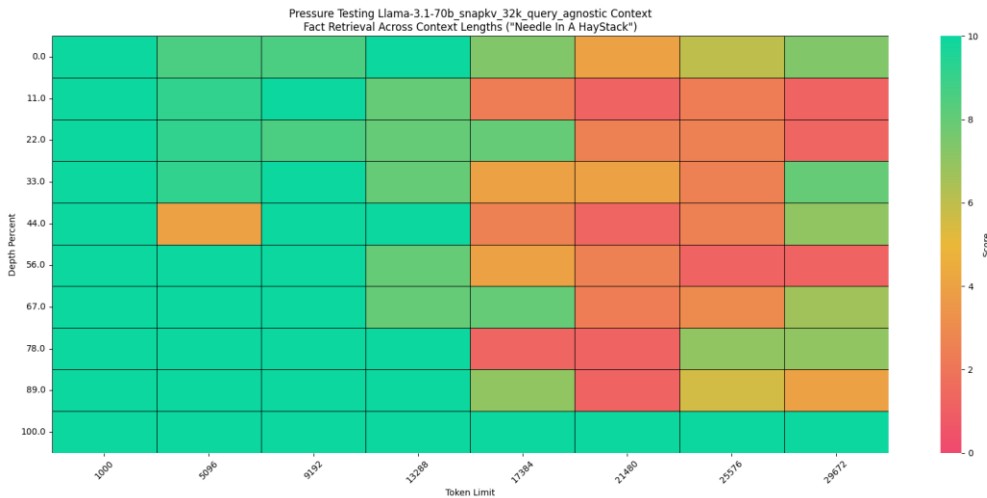

In contrast, RazorAttention maintains robust performance regardless of query availability with the same compression rate.

**Figure 3. RazorAttention(acc=98.33, max_length=128k)**

Pressure Testing Llama-3.1-70b_RA_128k Context
Fact Retrieval Across Context Lengths ("Needle In A HayStack")

# Section 2: Retrieval Head with Different Queries

The retrieval patterns remain highly consistent across different input queries, suggesting that they are indeed model-based rather than query-specific. In Figure 4, we illustrate the retrieval heads selected using various "Needle in a Haystack" queries, demonstrating that the selected heads are unchanged across different inputs.

**Figure 4. Retrieval Head with Different Queries**

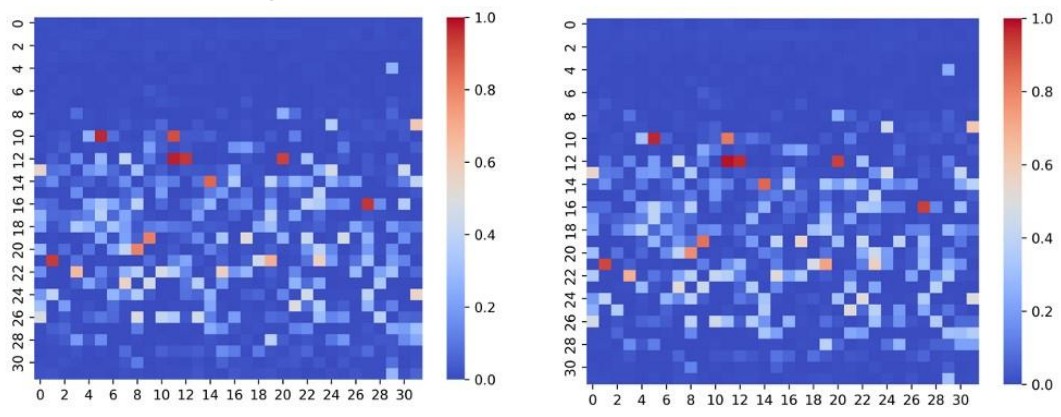