# OpenReview forum: "RazorAttention: Efficient KV Cache Compression Through Retrieval Heads"
_ICLR.cc/2025/Conference — ICLR 2025 Poster_

### Official Review · Reviewer_ZhVH · 2024-11-01

**Soundness:** 2
**Presentation:** 3
**Contribution:** 2
**Rating:** 5
**Confidence:** 4

**Summary:**

This paper investigates the attention mechanisms within transformer models, observing that most attention heads primarily focus on local context while only a few, termed "retrieval heads," are capable of attending to all input tokens. To optimize memory and computational efficiency, the authors propose maintaining a full key-value (kv) cache for retrieval heads and only keeping a recent kv cache for other attention heads. The identification of these retrieval heads is based on observed patterns known as the "Echo head" and "Induction head."

**Strengths:**

The paper is well-written and straightforward to follow.

**Weaknesses:**

1. The authors identify retrieval heads as echo heads and induction heads, demonstrated through an example using a synthetic repeated sequence. However, this example appears specifically designed to highlight these two types of heads. It raises the question of whether this pattern holds in more generalized query settings. Intuitively, it does not seem that in every type of query, the most crucial information is located solely in echo or induction tokens. Would these patterns still be dominant in more diverse and complex input data?

2. The authors state that they did not compare their method with SnapKV because SnapKV relies on the query for selecting the kv cache. However, this reasoning is weak, as SnapKV is a well-known and accepted method in the field. A comparison would provide more comprehensive insights.

3. I'm confusing that whether the selection of retrieval heads is model-based or  also query-specific. Are these retrieval heads consistent across different queries, or do different heads have different functions across different queries?

4. The authors note that reducing the kv cache for all non-retrieval heads introduces a notable accuracy gap and that compensation tokens can mitigate this gap. However, it is unclear whether the observed improvement comes purely from the use of compensation tokens or from the strategic allocation of the kv cache itself. An experiment comparing the combination of compensation tokens with other baseline models could clarify the extent of the improvement attributable to each factor.

**Questions:**

The paper lacks detailed experimental setup descriptions. When comparing with baseline methods, was the total kv budget (i.e., the combined kv cache size for all heads) kept constant?

---

> ### Author Response · Authors · 2024-11-24
>
> We sincerely appreciate your constructive feedback of our paper, and we will make certain updates w.r.t. your concerns in our revised version. Below are our responses to your concerns.
>
> Q1: Whether the retrieval pattern holds in general query settings.
>
> We appreciate this observation. Retrieval patterns remain highly consistent across diverse input queries, confirming they are predominantly model-based rather than query-specific. As detailed in Section 2 of our revised supplementary materials, we demonstrate the stability of retrieval head selection using varied “Needle in a Haystack” queries, where the identified heads remain unchanged across inputs.
>
> Further, we validate these findings on complex datasets such as LongBench and InfiniBench/Ruler (see results below), which encompass diverse and challenging task types. RazorAttention consistently retains the original performance. This evidence supports the robustness of the observed retrieval patterns.
>
> | InfiniteBench | Original Model | RA Model |
> |----------------|-----------------|-----------|
> | codedebug       | 21.57          | 21.57     |
> | ensum          | 30.71          | 30.58     |
> | endia          | 19.50          | 19.50     |
> | enqa           | 29.09          | 29.50     |
> | enmc           | 63.31          | 63.31     |
>
> | Ruler | Original Model | RA Model
> -- | -- | --
> ruler_niah_single_1_16k | 100 | 100
> ruler_niah_single_2_16k | 100 | 100
> ruler_niah_single_3_16k | 100 | 100
> ruler_niah_multikey_1_16k | 100 | 100
> ruler_niah_multikey_2_16k | 99.11 | 99.11
> ruler_niah_multikey_3_16k | 99.11 | 99.11
> ruler_niah_multivalue_16k | 99.11 | 97.54
> ruler_niah_multiquery_16k | 95.09 | 95.31
> ruler_vt_16k | 80.54 | 86.61
> ruler_fwe_16k | 89.29 | 85.42
> ruler_cwe_16k | 90.09 | 90.27
> ruler_qa_squad_16k | 88.39 | 88.39
> ruler_qa_hotpotqa_16k | 56.25 | 56.25
> ruler_16k | 92.07538462 | 92.15462
> ruler_niah_single_1_32k | 100 | 100
> ruler_niah_single_2_32k | 100 | 100
> ruler_niah_single_3_32k | 100 | 100
> ruler_niah_multikey_1_32k | 100 | 100
> ruler_niah_multikey_2_32k | 96.43 | 96.43
> ruler_niah_multikey_3_32k | 100 | 100
> ruler_niah_multivalue_32k | 97.54 | 94.2
> ruler_niah_multiquery_32k | 88.17 | 91.52
> ruler_vt_32k | 87.86 | 93.04
> ruler_fwe_32k | 88.69 | 88.99
> ruler_cwe_32k | 12.41 | 20.45
> ruler_qa_squad_32k | 86.61 | 85.71
> ruler_qa_hotpotqa_32k | 46.43 | 46.43
> ruler_32k | 84.93384615 | 85.90538
>
>
> Q2: Comparison with SnapKV.
>
> Thank you for this suggestion. SnapKV is indeed a prominent method for KV cache compression. Below, we provide a comparative analysis of RazorAttention and SnapKV in both query-aware and query-agnostic settings. While SnapKV excels in query-aware scenarios, its performance degrades significantly in query-agnostic tasks. RazorAttention’s training-free design and headwise sparse patterns ensure superior robustness across diverse queries.
> | RazorAttention   | SnapKV (query-aware)   |  SnapKV (query-agnostic)   |
> |------------|------------|------------|
> | 98.33 | 100 | 69.75 |
>
> Additionally, we include complete results for LongBench (Llama3.1-8B-instruct) to demonstrate the practical advantages of RazorAttention over SnapKV, especially under query-agnostic settings. This discussion will be incorporated into the revised paper.
>
> | Dataset                  | Baseline | RazorAttention | SnapKV |
> |--------------------------|----------|----------------|--------|
> | 2wikimqa                | 49.25    | 49.81          | 49.12  |
> | hotpotqa                | 57.61    | 57.22          | 57.60  |
> | musique                 | 33.72    | 32.80          | 32.55  |
> | multifieldqa_en         | 55.77    | 56.64          | 56.19  |
> | multifieldqa_zh         | 63.47    | 63.81          | 62.99  |
> | narrativeqa             | 29.23    | 29.54          | 30.03  |
> | qasper                  | 47.53    | 47.32          | 47.61  |
> | triviaqa                | 91.50    | 91.20          | 91.50  |
> | gov_report              | 34.58    | 33.08          | 32.97  |
> | qmsum                   | 25.27    | 25.10          | 25.37  |
> | vcsum                   | 17.28    | 16.99          | 16.49  |
> | dureader                | 34.88    | 31.91          | 31.64  |
> | lcc                     | 24.68    | 24.62          | 24.64  |
> | repobench-p             | 25.57    | 25.36          | 25.33  |
> | passage_retrieval_en    | 99.50    | 100.00         | 99.50  |
> | passage_retrieval_zh    | 90.45    | 95.98          | 90.45  |
> | passage_count           | 10.08    | 9.75           | 9.83   |
> | trec                    | 14.50    | 9.25           | 17.67  |
> | lsht                    | 0.00     | 0.00           | 0.00   |
> | multi_news              | 26.92    | 26.81          | 26.77  |
> | Samsum                  | 13.50    | 13.97          | 13.37  |

---

> ### Author Response · Authors · 2024-11-24
>
> Q3: The retrieval pattern among different queries.
>
> As clarified in Q1, the selection of retrieval heads is consistent across queries, indicating that these patterns are primarily model-driven rather than query-specific.
>
> Q4: The effectiveness of the compensation token.
>
> Our ablation studies (Figure 7 in the main paper) show that compensation tokens significantly improve accuracy, even with identical KV cache allocations. This evidence highlights the critical role of compensation tokens in mitigating information loss due to KV cache reduction. Exploring the combination of compensation tokens with other baseline algorithms presents an exciting direction for future work, and we aim to explore this in subsequent studies.
>
> Q5: The KV cache budget for different algorithms.
>
> To ensure fairness, all baseline methods were evaluated under the same total compression ratio, as detailed in Table 2 of the main paper. While the absolute KV cache budget may vary proportionally with input length due to RazorAttention’s dynamic compression strategy, the total compression ratio remains consistent across all methods for comparison purposes.

---

> > ### Author Response · Authors · 2024-11-25
> >
> > Dear reviewer, thank you once again for the time devoted to handling this work. Your insights have been helpful, and we sincerely appreciate your thoughtful review. Please don’t hesitate to let us know if there are any additional questions or points we can further clarify.

---

> > > ### Comment · Reviewer_ZhVH · 2024-11-26
> > >
> > > Thank you for your response. I will update my score to 5.
> > > However, I agree with Reviewers TBcp and 2S3S, and I believe the authors should compare their method with baselines under varying compression rates. If RazorAttention only outperforms the baselines at certain compression rates, its significance may be limited.

---

> > > > ### Author Response · Authors · 2024-12-03
> > > >
> > > > We sincerely appreciate your valuable feedback. In response, below we present the results of RazorAttention and SnapKV under 15%, 20%, and 40% KV cache budget for Needle In A Haystack. We use Llama3.1-70b and masked out the queries for both methods (to simulate the situation where the user queries are unknown).
> > > >
> > > > | Needle In A Haystack | RazorAttention | SnapKV |
> > > > |----------------|-----------------|-----------|
> > > > | 15%       | 100%         | 65%    |
> > > > | 20%         | 100%        | 69%    |
> > > > | 40%          |100%         | 78%     |

---

### Official Review · Reviewer_CNZf · 2024-11-03

**Soundness:** 3
**Presentation:** 3
**Contribution:** 2
**Rating:** 5
**Confidence:** 4

**Summary:**

The paper introduces RAZORATTENTION, motivated by the goal of reducing the size of the key-value cache as much as possible without losing semantic information. The rationale behind this approach is the retrieval heads in LLMs. For these retrieval heads, RAZORATTENTION retains the KV cache in its original state; for the other heads, it only caches recent tokens and attention sinks. Additionally, the paper designs a compensation strategy for compensating the information loss. Experiments indicate that RazorAttention can compress up to 70% of the KV cache without causing a noticeable decline in performance.

**Strengths:**

1. The research problem addressed in this paper is significant. Reducing the key-value  cache in large language models is a hot topic, and improvements in this area can have a substantial impact on the real-world deployment of LLMs.
2. Based on the phenomenon of retrieval heads present in large language models and in conjunction with the token dropping method, this paper explores an training-free way to reduce the key-value cache. The results from downstream tasks and NIH tests have both demonstrated the effectiveness of this approach.
3. The writing of this article is easy for readers to understand.

**Weaknesses:**

1. I agree that this work is the first training-free token reduction algorithm based on retrieval heads. However, to my knowledge, the phenomenon of retrieval heads being prevalent in LLMs was first articulated in [1]. [1] provides a detailed discussion on the properties of retrieval heads, which diminishes the novelty of this paper. Although the paper provides a theoretical analysis of AliBi-based models, which is commendable, it does not cover Rope-based models.
2. A recent work [2] is also motivated by the presence of retrieval heads. The main idea of [2] is to combine streaming heads with retrieval heads, which is very similar to the approach taken in this paper. But [2] is not training-free. It would be beneficial to elaborate on the advantages of this paper over [2].

[1] Retrieval head mechanistically explains long-context factuality.
[2] DuoAttention: Efficient Long-Context LLM Inference with Retrieval and Streaming Heads

**Questions:**

1. As mentioned in the weaknesses, it is beneficial to compare with DuoAttention in terms of effectiveness.
2. As a training-free method, a potential issue is that different models may have different optimal hyperparameters. For example, how can we quickly determine the suitable proportion of Induction Heads and Echo Heads for different models?
3. If a general theoretical analysis can be found explaining why retrieval heads are widely present in various models, that would be highly significant.

---

> ### Author Response · Authors · 2024-11-24
>
> We sincerely appreciate your constructive feedback of our paper, and we will make certain updates w.r.t. your concerns in our revised version. Below are our responses to your concerns.
>
> Q1: Comparison with [1].
>
> Although [1] does provide a similar finding as ours, directly retaining the KV cache in induction heads introduces a significant accuracy drop for tasks such as Needle In A Haystack. We introduce two essential strategies that go beyond the use of induction heads to achieve lossless performance:
> 1. Expanded Definition of Retrieval Heads:
> The concept of induction heads was initially introduced in [1], where these heads were identified as attention heads that follow a token-retrieval pattern (attending to previously seen tokens). In [2], the authors extended this observation to long-context scenarios, showing that induction heads are more critical than other heads under such conditions. However, these works considered only induction heads as retrieval heads. Our analysis reveals that echo heads are equally crucial for retaining model performance. For instance, when tested on the Needle in a Haystack dataset, using the definition of retrieval heads from [1] leads to an approximate 10% accuracy drop. By redefining retrieval heads to include both induction and copy heads, we retain the full performance of the model (see Figure 5 in our paper).
> 2. Compensation Token Strategy:
> Directly discarding remote tokens in non-retrieval heads results in severe performance degradation, with accuracy drops exceeding 30%. To address this, we designed the compensation token strategy, which condenses the dropped information from these heads into a compact form. This strategy ensures that essential information is preserved while enabling efficient KV cache compression.
>
> Q2: Comparison with [2].
>
> An obvious advantage of the training-free algorithm is that it can be used as a plug-and-play component for advanced LLM serving system, such as TRT-LLM and Triton, and training-free lossless compression is usually a much more challenging task compared to the training-based algorithms. Moreover, we believe the compensation token we introduced and our head selection method (inductive + echo heads) provide valuable insights into the underlying functionality of LLMs in long-context scenarios.
>
> Q3: How to quickly determine the suitable proportion of Induction Heads and Echo Heads for different models.
>
> We find that for most open-source models, about 15%~20% induction heads+1% echo heads shall be enough without training. However, if there still exists certain performance loss, our recipe is to increase about 3% of the induction heads at one step. We will provide a discussion section about this in our revised version.
>
> Q4: General theoretical analysis explaining why retrieval heads are widely present in various models.
>
> We believe this is a quite important and challenging topic, and we will leave this for future study.

---

> > ### Comment · Reviewer_CNZf · 2024-11-25
> >
> > Thank you for your rebuttal. However, I will keep my rating unchanged.

---

### Official Review · Reviewer_gP7k · 2024-11-04

**Soundness:** 3
**Presentation:** 2
**Contribution:** 2
**Rating:** 5
**Confidence:** 4

**Summary:**

This paper introduce a KV cache compression algorithm in LLMs that only maintains a full cache for retrieval heads and reduces all the remote tokens in non-retrieval heads to single compensation token per head. The proposed method can achieve 3X KV cache compression ratio without noticeable impacts on performance.

**Strengths:**

1.	It is pluggable solution that enhances the inference efficiency of a diverse set of large language models without noticeable impacts on performance.

**Weaknesses:**

1.	There are some related works identifying the different patterns of retrieval and non-retrieval heads, such as DuoAttention, Retrieval Head Mechanistically Explains Long-Context Factuality, MInference, etc. It would be better to clarify the novelty of the proposed method.
2.	It can only be applied to multiple KV cache scenarios. For only single KV cache model structure, such as YOCO, it cannot be applied.
3.	The baselines are not comprehensive. It would be beneficial if the authors could compare their results with more advanced baselines.
4.	It seems the proposed method cannot achieve the best performance in all the benchmarks. It would be beneficial to give detailed analysis of the worse cases.

**Questions:**

1.	What is the compression ratio for GQA in Llama3-8B-Instruct model? What is the compression ratio for 1024K sequence length?
2.	How to decide the 1% echo heads and 14% induction heads configuration? Have the authors tried different combination, e.g. more echo heads and less induction heads?

---

> ### Author Response · Authors · 2024-11-24
>
> We sincerely appreciate your constructive feedback of our paper, and we will make certain updates w.r.t. your concerns in our revised version. Below are our responses to your concerns.
>
> Q1: Novelty of RazorAttention.
>
> Our work is the first to apply induction heads for KV cache compression, and we introduce two essential strategies that go beyond the use of induction heads to achieve lossless performance:
> 1. Expanded Definition of Retrieval Heads:
> The concept of induction heads was initially introduced in [1], where these heads were identified as attention heads that follow a token-retrieval pattern (attending to previously seen tokens). In [2], the authors extended this observation to long-context scenarios, showing that induction heads are more critical than other heads under such conditions. However, these works considered only induction heads as retrieval heads. Our analysis reveals that echo heads are equally crucial for retaining model performance. For instance, when tested on the Needle in a Haystack dataset, using the definition of retrieval heads from [1] leads to an approximate 10% accuracy drop. By redefining retrieval heads to include both induction and copy heads, we retain the full performance of the model (see Figure 5 in our paper).
> 2. Compensation Token Strategy:
> Directly discarding remote tokens in non-retrieval heads results in severe performance degradation, with accuracy drops exceeding 30%. To address this, we designed the compensation token strategy, which condenses the dropped information from these heads into a compact form. This strategy ensures that essential information is preserved while enabling efficient KV cache compression.
>
> Q2: More comparison.
>
> According to your suggestion, here we also include the comparison of RazorAttention with Snapkv at the same compression rate, under both query-aware and query-agnostic settings. While SnapKV excels in query-aware scenarios, its performance degrades significantly in query-agnostic tasks. RazorAttention’s training-free design and headwise sparse patterns ensure superior robustness across diverse queries.
> | RazorAttention   | SnapKV (query-aware)   |  SnapKV (query-agnostic)   |
> |------------|------------|------------|
> | 98.33 | 100 | 69.75 |
>
> Additionally, we include complete results for LongBench (Llama3.1-8B-instruct) to demonstrate the practical advantages of RazorAttention over SnapKV, especially under query-agnostic settings. This discussion will be incorporated into the revised paper.
>
> | Dataset                  | Baseline | RazorAttention | SnapKV |
> |--------------------------|----------|----------------|--------|
> | 2wikimqa                | 49.25    | 49.81          | 49.12  |
> | hotpotqa                | 57.61    | 57.22          | 57.60  |
> | musique                 | 33.72    | 32.80          | 32.55  |
> | multifieldqa_en         | 55.77    | 56.64          | 56.19  |
> | multifieldqa_zh         | 63.47    | 63.81          | 62.99  |
> | narrativeqa             | 29.23    | 29.54          | 30.03  |
> | qasper                  | 47.53    | 47.32          | 47.61  |
> | triviaqa                | 91.50    | 91.20          | 91.50  |
> | gov_report              | 34.58    | 33.08          | 32.97  |
> | qmsum                   | 25.27    | 25.10          | 25.37  |
> | vcsum                   | 17.28    | 16.99          | 16.49  |
> | dureader                | 34.88    | 31.91          | 31.64  |
> | lcc                     | 24.68    | 24.62          | 24.64  |
> | repobench-p             | 25.57    | 25.36          | 25.33  |
> | passage_retrieval_en    | 99.50    | 100.00         | 99.50  |
> | passage_retrieval_zh    | 90.45    | 95.98          | 90.45  |
> | passage_count           | 10.08    | 9.75           | 9.83   |
> | trec                    | 14.50    | 9.25           | 17.67  |
> | lsht                    | 0.00     | 0.00           | 0.00   |
> | multi_news              | 26.92    | 26.81          | 26.77  |
> | Samsum                  | 13.50    | 13.97          | 13.37  |
>
>
>   LongBench | Baseline | RazorAttention | SnapKV
> -- | -- | -- | --
> 2wikimqa | 49.25 | 49.81 | 49.12
> hotpotqa | 57.61 | 57.22 | 57.6
> musique | 33.72 | 32.8 | 32.55
> multifieldqa_en | 55.77 | 56.64 | 56.19
> multifieldqa_zh | 63.47 | 63.81 | 62.99
> narrativeqa | 29.23 | 29.54 | 30.03
> qasper | 47.53 | 47.32 | 47.61
> triviaqa | 91.5 | 91.2 | 91.5
> gov_report | 34.58 | 33.08 | 32.97
> qmsum | 25.27 | 25.1 | 25.37
> vcsum | 17.28 | 16.99 | 16.49
> dureader | 34.88 | 31.91 | 31.64
> lcc | 24.68 | 24.62 | 24.64
> repobench-p | 25.57 | 25.36 | 25.33
> passage_retrieval_en | 99.5 | 100 | 99.5
> passage_retrieval_zh | 90.45 | 95.98 | 90.45
> passage_count | 10.08 | 9.75 | 9.83
> trec | 14.5 | 9.25 | 17.67
> lsht | 0 | 0 | 0
> multi_news | 26.92 | 26.81 | 26.77
> Samsum | 13.5 | 13.97 | 13.37

---

> > ### Author Response · Authors · 2024-11-24
> >
> > Q3: Analysis of the worse cases.
> >
> > We appreciate your observation regarding the benchmarks where RazorAttention does not achieve the best performance.
> > From our analysis, we found that most of the challenging cases occur in summarization tasks. The performance loss in these tasks can be attributed to the model’s tendency to produce shorter answers when using RazorAttention. Since summarization tasks often rely on metrics like F1 score, which heavily penalize shorter responses, this behavior leads to lower scores.
> >
> > Q4: What is the compression ratio for GQA in Llama3-8B-Instruct model.
> >
> > For GQA models, we directly select 15% of the groups in the model (the score of each group is the sum of the heads within the group), therefore the compression ratio is the same with MHA models.
> >
> > Q5: Different combinations of the echo heads and induction heads.
> >
> > Yes, the performance of RazorAttention under different heads budget can be found in Figure.5 and Table.4 of our paper. Basically we noticed that including only 1% of the echo heads can essentially improve the accuracy while we need about 15% induction heads to fully recover the performance of the original model.

---

> > > ### Author Response · Authors · 2024-11-25
> > >
> > > Dear reviewer, thank you once again for the time devoted to handling this work. Your insights have been helpful, and we sincerely appreciate your thoughtful review. Please don’t hesitate to let us know if there are any additional questions or points we can further clarify.

---

> > > > ### Comment · Reviewer_gP7k · 2024-12-03
> > > >
> > > > Thank you for your detailed response to my review. Regarding to the concern of Q1, Q2 and Q3, the paper could be further improved. Therefore, I will keep my score unchanged.

---

### Official Review · Reviewer_2S3S · 2024-11-04

**Soundness:** 3
**Presentation:** 3
**Contribution:** 3
**Rating:** 6
**Confidence:** 4

**Summary:**

The paper introduces RazorAttention, a training-free algorithm for compressing KV cach in LLMs. This method selectively retains information in specific “retrieval heads” of the multi-head attention mechanism while discarding remote tokens in other heads. A notable aspect of RazorAttention is using a "compensation token" to summarize discarded information, minimizing data loss. RazorAttention shows a 70% reduction in KV cache size with minimal performance impact, supporting compatibility with FlashAttention for efficient large-context inference without modifying the base model.

**Strengths:**

- RazorAttention’s head-specific strategy, distinguishing retrieval heads from non-retrieval heads, is a novel approach to KV cache compression that aligns with the attention heads’ different roles.
- This method is compatible with FlashAttention and offers practical efficiency gains without training or model alterations.
- RazorAttention is tested across various architectures, including ALiBi and RoPE models, proving effective on different LLM types, with compression ratios of up to 70%.

**Weaknesses:**

- The paper lacks detailed efficiency reports, particularly on latency and throughput improvements. A breakdown of the actual performance gains would give better insight into its practical benefits.
- RazorAttention introduces operations like compensation token creation and slicing, yet the associated computational overhead is not profiled and discussed. A detailed overhead analysis and implementation would help.
- RazorAttention’s comparison is primarily with StreamingLLM and H2O, omitting comparison and discussion of other types of KV cache compression methods. (like what are others’ performances under similar compression rates? can this method be combined with others?)
- The results focus on one compression rate without experiments on varying compression ratios compared with others.

**Questions:**

- What are the latency and throughput improvements, both absolute and relative to the baseline?
- What are the comparative performances at different KV cache compression ratios?
- Has RazorAttention been tested on other models like Mistral-7B-Instruct-v0.2 or longchat-7b-v1.5-32k?
- Can you provide a breakdown or approximation of the distribution between retrieval and non-retrieval heads?

---

> ### Author Response · Authors · 2024-11-24
>
> We sincerely appreciate your constructive feedback of our paper, and we will make certain updates w.r.t. your concerns in our revised version. Below are our responses to your concerns.
>
> Q1: The efficiency evaluation.
>
> We evaluate RazorAttention’s decoding latency and throughput on GLM-9B-1M model on 8 Ascend 910B NPUs under different input lengths for both prefill and decoding
>
> | Input Length | Batch Size | Prefill Speedup | Decoding Speedup |
> |--------------|------------|-----------------|------------------|
> | 128k         | 1          | 1.2%          | 3.1%           |
> |              | 4          | 5.66%           | 15.9%            |
> |              | 8          | 8.03%           | 26.67%           |
> |              | 10         | 7.05%           | 29.47%           |
> |              | 18         | From OOM to feasible inference |              |
> | 256k         | 1          | 6.58%           | 9.2%           |
> |              | 4          | 8.76%           | 25.47%           |
> |              | 5          | 9.93%           | 29.01%           |
> |              | 9          | From OOM to feasible inference |              |
>
> Below is the maximum throughput improvement of using RazorAttention under different input lengths:
>
> | Input Length | Maximum Throughput Improvement |
> |--------------|------------|
> |128k | 64.2% |
> |256k | 71.81% |
>
>
> Q2: Overhead Analysis.
>
>  After computing the KV Cache at each layer, we directly calculate the average of the tokens that need to be compressed. Only the averaging operation is introduced, and the capacity of the compressed KV Cache is significantly reduced. When transferring the data back from the compute unit to the storage unit, the data transfer overhead is also reduced. In summary, this operation does not introduce significant performance overhead, so our overall performance improves in long-sequence scenarios for the prefill stage as well.
>
> Q3: Comparison with other algorithms.
>
> Below we evaluated the accuracy of Snapkv at the same compression rate using LongBench on the Llama3.1-8B-instruct model.
>
> | Dataset                  | Baseline | RazorAttention | SnapKV |
> |--------------------------|----------|----------------|--------|
> | 2wikimqa                | 49.25    | 49.81          | 49.12  |
> | hotpotqa                | 57.61    | 57.22          | 57.60  |
> | musique                 | 33.72    | 32.80          | 32.55  |
> | multifieldqa_en         | 55.77    | 56.64          | 56.19  |
> | multifieldqa_zh         | 63.47    | 63.81          | 62.99  |
> | narrativeqa             | 29.23    | 29.54          | 30.03  |
> | qasper                  | 47.53    | 47.32          | 47.61  |
> | triviaqa                | 91.50    | 91.20          | 91.50  |
> | gov_report              | 34.58    | 33.08          | 32.97  |
> | qmsum                   | 25.27    | 25.10          | 25.37  |
> | vcsum                   | 17.28    | 16.99          | 16.49  |
> | dureader                | 34.88    | 31.91          | 31.64  |
> | lcc                     | 24.68    | 24.62          | 24.64  |
> | repobench-p             | 25.57    | 25.36          | 25.33  |
> | passage_retrieval_en    | 99.50    | 100.00         | 99.50  |
> | passage_retrieval_zh    | 90.45    | 95.98          | 90.45  |
> | passage_count           | 10.08    | 9.75           | 9.83   |
> | trec                    | 14.50    | 9.25           | 17.67  |
> | lsht                    | 0.00     | 0.00           | 0.00   |
> | multi_news              | 26.92    | 26.81          | 26.77  |
> | Samsum                  | 13.50    | 13.97          | 13.37  |
>
> However, when the future queries are unknown for SnapKV, its performance drops significantly. Below, we present the results of SnapKV in both query-aware and query-agnostic settings for Needle In A Haystack: (see detailed Figures in Section.1 of our supplementary updated):
> | RazorAttention   | SnapKV (query-aware)   |  SnapKV (query-agnostic)   |
> |------------|------------|------------|
> | 98.33 | 100 | 69.75 |
>
> Only RazorAttention is still feasible in this case. This resilience is due to its training-free design and headwise sparse pattern, which ensure minimal information loss across various query types.
>
> Our algorithm can be perfectly combined with current token-dropping algorithms, such as H2O, SnapKV, MInference. Since we only constrain the cache budget in the short heads while in long heads the KV cache can be further compressed using these algorithms.
>
> Q4: Results of RazorAttention under different compression ratios.
>
> The reason why we only consider one specific ratio is that we found the ratio of the necessary retrieval heads is about 15%. Using fewer retrieval heads would introduce a certain accuracy drop (see Table.4 in our paper) while using more heads gives no more accuracy gain. Therefore we only report the comparison result under this compression ratio for all baselines. We will clarify this in our revised version.

---

> > ### Author Response · Authors · 2024-11-24
> >
> > Q5: Breakdown of the distribution between retrieval and non-retrieval heads.
> >
> > According to your suggestion, we plot the distribution of the retrieval and non-retrieval heads in Section.2 of our updated supplementary. We notice a clear trend in which most retrieval heads distribute in the middle depth of the network.
> >
> > Q6. More results for other models.
> >
> > Due to the time limit, we plan to add the results for longchat-7b-v1.5-32k in our revised version.

---

> ### Comment · Reviewer_2S3S · 2024-11-25
>
> Thanks for the answers to my questions; I've increased the rating. I have some follow-ups here:
> - For the efficiency tests, please also use GPUs like A100, H100, etc.
> - I would like to see performance when combining with other methods like token-dropping, quantization, etc. To see how far can we go in terms of KV cache compression.
> - For figure 4 in the supplement, what is the model? And can you further elaborate on how to interpret the result and what heads are retrieval heads?

---

> > ### Author Response · Authors · 2024-12-03
> >
> > We sincerely appreciate your valuable feedback. In response, we will incorporate the efficiency results tested on A100  in the updated version of our manuscript.
> > Below we further compress the KV cache under RazorAttention into MX-FP4 on Llama3.1-8B and report its results on LongBench.
> >
> > | Dataset                  | baseline | KV MXFP4+RA |
> > |--------------------------|----------|-------------|
> > | 2wikimqa                 | 49.25    | 47.64       |
> > | hotpotqa                 | 57.61    | 55.26       |
> > | musique                  | 32.72    | 30.7        |
> > | multifieldqa_en          | 55.77    | 57.23       |
> > | multifieldqa_zh          | 63.47    | 61.2        |
> > | narrativeqa              | 29.23    | 30.32       |
> > | qasper                   | 47.53    | 47.09       |
> > | triviaqa                 | 91.5     | 91.79       |
> > | gov_report               | 34.58    | 33.57       |
> > | qmsum                    | 25.27    | 25.41       |
> > | vcsum                    | 17.28    | 17.29       |
> > | dureader                 | 34.88    | 33.62       |
> > | lcc                      | 24.63    | 23.74       |
> > | repobench-p              | 25.57    | 23.66       |
> > | passage_retrieval_en     | 99.5     | 100         |
> > | passage_retrieval_zh     | 90.45    | 95.22       |
> > | passage_count            | 10.08    | 5.64        |
> > | trec                     | 14.5     | 13.04       |
> > | lsht                     | 0        | 0.5         |
> > | multi_news               | 26.92    | 26.66       |
> > | Samsum                   | 13.5     | 13.23       |
> >
> > For Figure 4 in the supplement, we employed Qwen1.5-7B. In this figure, the y-axis represents the layer number, and the x-axis corresponds to the head index. A higher score in the heatmap indicates that the corresponding attention head achieves a higher retrieval score. Based on these scores, we selected the top-x% of attention heads as the retrieval heads.

---

### Official Review · Reviewer_hKDU · 2024-11-05

**Soundness:** 3
**Presentation:** 3
**Contribution:** 2
**Rating:** 6
**Confidence:** 5

**Summary:**

The paper has made an observation: induction and echo heads are retrieval heads and others are local processing heads. They then keep complete KV Cache for retrieval heads and drop non-local KV for local processing heads. Following their hyper parameters, this leads to 70% asymptotic KV Compression.

**Strengths:**

1. The observation is neat.  And experiments support the hypothesis quite well.
2. The recipe to identify the heads is very lightweight (although is this a contribution or is it previously known from anthropic paper?)

**Weaknesses:**

1. Experiments

     A. why are some datasets from longbench missing like passage_retrieval etc

     B. Can you perform experiments on infinity benchmark for comparison.

     C. what is the exact KV compression in the longbench datasets? The 70% number is misleading since longbench generally has small context lengths and authors have used a context thold of 4000.  Please add exact KV Cache compression numbers for all the methods to the table.

2. Novelty: Can you please elaborate on the novelty of defining / identifying the echo and induction heads. what is already known and what is figured out?

3. Writing errors: Please revise the manuscript for text/language errors. e.g. :

     A. "There are been plenty ..." (page 1)

     B. " and proved that accuracy .." ( page 2) {you showed it empirically .. and also 70% is not what you show in the experiments.}.

     C. C_0 in equation 2. Do you mean L_h ?

**Questions:**

Please address the weaknesses above.

---

> ### Author Response · Authors · 2024-11-24
>
> We sincerely appreciate your constructive feedback of our paper, and we will make certain updates w.r.t. your concerns in our revised version. Below are our responses to your concerns.
>
> Q1: More experimental results.
>
> As you suggest, we have included the results of infinite bench  and ruler below with Llama3.1-8B-instruct under the same setting of our paper. Due to time constraints, only a subset of the experiments have been included. We will present additional results upon completion of the remaining experiments.
> | InfiniteBench | Original Model | RA Model |
> |----------------|-----------------|-----------|
> | codedebug       | 21.57          | 21.57     |
> | ensum          | 30.71          | 30.58     |
> | endia          | 19.50          | 19.50     |
> | enqa           | 29.09          | 29.50     |
> | enmc           | 63.31          | 63.31     |
>
>  Ruler | Original Model | RA Model
> -- | -- | --
> ruler_niah_single_1_16k | 100 | 100
> ruler_niah_single_2_16k | 100 | 100
> ruler_niah_single_3_16k | 100 | 100
> ruler_niah_multikey_1_16k | 100 | 100
> ruler_niah_multikey_2_16k | 99.11 | 99.11
> ruler_niah_multikey_3_16k | 99.11 | 99.11
> ruler_niah_multivalue_16k | 99.11 | 97.54
> ruler_niah_multiquery_16k | 95.09 | 95.31
> ruler_vt_16k | 80.54 | 86.61
> ruler_fwe_16k | 89.29 | 85.42
> ruler_cwe_16k | 90.09 | 90.27
> ruler_qa_squad_16k | 88.39 | 88.39
> ruler_qa_hotpotqa_16k | 56.25 | 56.25
> ruler_16k | 92.07538462 | 92.15462
> ruler_niah_single_1_32k | 100 | 100
> ruler_niah_single_2_32k | 100 | 100
> ruler_niah_single_3_32k | 100 | 100
> ruler_niah_multikey_1_32k | 100 | 100
> ruler_niah_multikey_2_32k | 96.43 | 96.43
> ruler_niah_multikey_3_32k | 100 | 100
> ruler_niah_multivalue_32k | 97.54 | 94.2
> ruler_niah_multiquery_32k | 88.17 | 91.52
> ruler_vt_32k | 87.86 | 93.04
> ruler_fwe_32k | 88.69 | 88.99
> ruler_cwe_32k | 12.41 | 20.45
> ruler_qa_squad_32k | 86.61 | 85.71
> ruler_qa_hotpotqa_32k | 46.43 | 46.43
> ruler_32k | 84.93384615 | 85.90538
>
> Meanwhile, here we include the full results of RazorAttention on Longbench with Llama3.1-8B-instruct .
> | Dataset               | Baseline | RazorAttention |
> |-----------------------|----------|----------------|
> | 2wikimqa             | 49.25    | 49.81          |
> | hotpotqa             | 57.61    | 57.22          |
> | musique              | 33.72    | 32.80          |
> | multifieldqa_en      | 55.77    | 56.64          |
> | multifieldqa_zh      | 63.47    | 63.81          |
> | narrativeqa          | 29.23    | 29.54          |
> | qasper               | 47.53    | 47.32          |
> | triviaqa             | 91.50    | 91.20          |
> | gov_report           | 34.58    | 33.08          |
> | qmsum                | 25.27    | 25.10          |
> | vcsum                | 17.28    | 16.99          |
> | dureader             | 34.88    | 31.91          |
> | lcc                  | 24.68    | 24.62          |
> | repobench-p          | 25.57    | 25.36          |
> | passage_retrieval_en | 99.50    | 100.00         |
> | passage_retrieval_zh | 90.45    | 95.98          |
> | passage_count        | 10.08    | 9.75           |
> | trec                 | 14.50    | 9.25           |
> | lsht                 | 0.00     | 0.00           |
> | multi_news           | 26.92    | 26.81          |
> | Samsum               | 13.50    | 13.97          |
>
>
> Q2: The exact compression ratio.
>
> We compress 85% of the non-retrieval heads, with the compressed size set to max(4000, N/5), where \( N \) represents the length of the input sequence. The compression ratio on each subset of LoneBench is correlated with the dataset's sequence length, and we ensured a fair comparison for H2O by setting the KV cache budget to match that of RazorAttention for each sentence.

---

> > ### Author Response · Authors · 2024-11-24
> >
> > Q3: Novelty of RazorAttention compared to work related.
> >
> > We are the first work that applies the induction heads for KV cache compression, and we introduce two essential strategies beyond the use of induction heads to achieve lossless performance. The concept of induction heads was initially proposed in [1], where the authors observed that specific attention heads exhibit a retrieval pattern by attending to previously existing tokens. Recent work [2] extended this observation to long-context scenarios, demonstrating through ablation studies that induction heads are more critical than other heads under such conditions.
> > Our work builds on these insights and contributes two major advancements:
> > 1. Redefining Retrieval Heads:
> > Unlike [2], which considers only induction heads as retrieval heads, our analysis identifies that echo heads are equally crucial for preserving performance. For example, when tested on the Needle in a Haystack dataset, using the definition of retrieval heads from [2] leads to a ~10% accuracy drop. By expanding the definition to include copy heads, we ensure comprehensive retention of the model’s capabilities (as shown in Figure 5 of our paper).
> > 2. Designing the Compensation Token Strategy:
> > Directly discarding remote tokens in non-retrieval heads results in significant performance degradation, with accuracy drops exceeding 30%. To address this, we introduce a novel compensation token strategy that condenses the information from discarded tokens. This innovation preserves essential information, mitigating the adverse effects of token removal and enabling efficient KV cache compression without compromising performance.
> >
> > Q4: Writing Errors.
> >
> > Thank you for pointing out the textual and linguistic issues. We will revise the manuscript thoroughly to address these errors and improve overall clarity.
> >
> > [1]. In-context Learning and Induction Heads
> > [2]. Retrieval Head Mechanistically Explains Long-Context Factuality

---

> > > ### Author Response · Authors · 2024-11-25
> > >
> > > Dear reviewer, thank you once again for the time devoted to handling this work. Your insights have been helpful, and we sincerely appreciate your thoughtful review. Please don’t hesitate to let us know if there are any additional questions or points we can further clarify.

---

> > > > ### Comment · Reviewer_hKDU · 2024-11-27
> > > >
> > > > The additional results alleviate my concerns. I am raising the score.

---

> > > > > ### Author Response · Authors · 2024-12-03
> > > > >
> > > > > Thank you for your hard work and recognition of our paper. We're glad to hear that you're satisfied.

---

### Official Review · Reviewer_TBcp · 2024-11-06

**Soundness:** 3
**Presentation:** 2
**Contribution:** 2
**Rating:** 5
**Confidence:** 4

**Summary:**

The paper proposed a novel KV cache mechanism basing on retrieval heads. To be more specific, the paper presents the observation of some retrieval heads that is essential for handling long context workload, then proposed a token-dropping based method to compress the to KV cache of the non-retrieval heads with compensation tokens, while keeping the retrieval head integrated, leading to substantial accuracy improvement compare to previous proposed method.

**Strengths:**

The paper observes the phenomenon of "retrieval heads" aligning with observations from other works.

Basing on the retrieval heads, the paper presents a straightforward token dropping based KV compression mechanism.

Experiments results show improvement compare with the previous token dropping based baselines.

**Weaknesses:**

Lack of experimental results on compression ratio:
The paper compares against StreamLLM and H2O. However, StreamLLM has a very limited compression ratio, and the performance of different compression ratio in H2O also will be different. The current manuscript lack the study of how different compression ratio in RazorAttention will affect the performance.

Lack of overhead evaluation on efficiency
The paper claims that RazorAttention can enhance LLM inference efficiency without overhead. However, there is no efficiency evaluation (both peak memory usage as well as the inference latency analysis). It is essential to have such real efficiency gain evaluation.

The novelty of "retrieval heads"
Previous paper [1] has demonstrated the existence of "retrieval heads" and its importance in serving long context. Do the authors aware of this and is there a difference between the two "retrieval heads" proposed in the two paper?

Miss of clarification:
"Retrieve and Process" in line 84: The manuscript claims that some heads are used for retrieval while some are used for process. However, lacks the analysis of "process"
"Protect" in line 234: what is the specific protection mechanism?

Potential typos:
115 as follows:

[1] Retrieval Head Mechanistically Explains Long-Context Factuality

**Questions:**

See the weakness

---

> ### Author Response · Authors · 2024-11-24
>
> We sincerely appreciate your constructive feedback of our paper, and we will make certain updates w.r.t. your concerns in our revised version. Below are our responses to your concerns.
>
> Q1: Clarify the experimental setup for H2O.
>
> For H2O, we ensured a fair comparison by setting the KV cache budget to match that of RazorAttention for each sentence. We appreciate your suggestion and will clarify this in our revised version.
>
> Q2: Efficiency evaluation results.
>
> We evaluate RazorAttention’s decoding latency and throughput on GLM-9B-1M model on 8 Ascend 910B NPUs under different input lengths for both prefill and decoding
>
> | Input Length | Batch Size | Prefill Speedup | Decoding Speedup |
> |--------------|------------|-----------------|------------------|
> | 128k         | 1          | 1.2%          | 3.1%           |
> |              | 4          | 5.66%           | 15.9%            |
> |              | 8          | 8.03%           | 26.67%           |
> |              | 10         | 7.05%           | 29.47%           |
> |              | 18         | From OOM to feasible inference |              |
> | 256k         | 1          | 6.58%           | 9.2%           |
> |              | 4          | 8.76%           | 25.47%           |
> |              | 5          | 9.93%           | 29.01%           |
> |              | 9          | From OOM to feasible inference |              |
>
> Below is the maximum throughput improvement of using RazorAttention under different input length:
>
> | Input Length | Maximum Throughput Improvement |
> |--------------|------------|
> |128k | 64.2% |
> |256k | 71.81% |
>
>
> Q3: Comparison with previous work [1].
>
> We acknowledge the similarities with [1] and highlight the following key distinctions:
> 1. Broader Definition of Retrieval Heads: In [1], the authors define retrieval heads as only the induction heads. In contrast, our analysis shows that beyond the inductive heads, about 1% of echo heads play an equally essential role in retaining model performance (refer to Figure 5 in our paper). Using the narrower definition in [1] results in an approximate 10% accuracy drop in the Needle In A Haystack benchmark, emphasizing the importance of our algorithm design.
> 2. Compensation Token Design: Directly applying a truncation on the non-retrieval heads' KV cache introduces a significant accuracy drop (>30%). Our compensation token design mitigates this issue effectively and ensures minimal information loss. In summary, while [1] provides foundational insights into retrieval heads, our work extends these findings to achieve lossless KV cache compression. We will add a detailed discussion of these distinctions in the revised version.
>
> Q4: Elaborate on the “Retrieve and Process” hypothesis.
>
> The “Retrieve and Process” hypothesis originates from the observed behavior of attention heads in LLMs. Most attention heads focus primarily on recent tokens, while only a few retrieval heads attend to remote tokens to extract relevant information. This mirrors the human cognitive process: recalling pertinent information before generating a response. We will clarify this point in our revised version.
>
> Q5: What does "Protect" mean.
>
>  In line 234, “protect” refers to preserving the KV cache in selected heads while discarding remote tokens in others.

---

> > ### Author Response · Authors · 2024-11-25
> >
> > Dear reviewer, thank you once again for the time devoted to handling this work. Your insights have been helpful, and we sincerely appreciate your thoughtful review. Please don’t hesitate to let us know if there are any additional questions or points we can further clarify.

---

> > > ### Comment · Reviewer_TBcp · 2024-12-02
> > > **Thanks for the rebuttal**
> > >
> > > I would like to thank the author for the rebuttal. I read other reviewer's comment and your interaction. The response of the efficiency evaluation looks good, and I hope more comprehensive and thorough evaluation can be done on direct efficiency measurement. For other writing and claiming perspectives, please make them clear in the future version of the paper. In light of the rebuttal, I raised my score.

---

> > > > ### Author Response · Authors · 2024-12-03
> > > >
> > > > We sincerely appreciate your valuable feedback. In response, we will incorporate the efficiency results tested on A100, using OpenAI Triton to implement this hybrid attention together with the compensation token, in our updated manuscript.

---

### Official Review · Reviewer_jpBx · 2024-11-13

**Soundness:** 3
**Presentation:** 2
**Contribution:** 3
**Rating:** 8
**Confidence:** 5

**Summary:**

This work proposed a method advocating keeping a full KV cache for retrieval/induction heads while adopting a StreamingLLM-like cache pattern for the rests. The proposed method claims to have strong long context performance and many efficiency perks.

**Strengths:**

1. The proposed method demonstrates decent performance, providing a significant improvement over many token dropping-based baselines.
2. The evaluation is thorough in terms of model coverages.
3. As recognized by many recent arts, being FlashAttention compatible is a prerequisite of being a pratical long context serving method, and this work fulfills this prerequisite.

**Weaknesses:**

1. The baseline methods (H2O and StreamingLLM) are dated and do not reflect the current SOTA of KV cache compression advancement. I understand if the authors do not intend to feature methods like KIVI or TOVA — as they are with different schools of approaches — but compare with sparse inference techniques like SnapKV or MInference are necessary.

The authors' discussion around L358 regarding SnapKV is faithful, as it corresponds to the findings of SharedContextBench (also submitted to ICLR here). However, SnapKV — and, in essence, methods like LoCoCo and LESS — are still worth featuring since we can almost always find weaknesses in any method. The authors should consider adding some quantifiable evaluations to reflect the weakness of SnapKV-like methods, if they intend to highlight the superiority of RazerAttention in this regard.

2. The needle setting is unclear. It is often very easy to do the "eating sandwich in a park" needle with the "grass sky sun" repetition background. But it is hard with the "magic city number" needle with a noisy background (e.g., Paul Graham's essay). The authors should clarify this setting and consider adding more standardized needle evaluations, like the RULER.

3. Similar to #2, the LongBench truncation settings of Table 3 can also use some clarifications.

4. RazorAttention is proposed as an efficiency work. However, the efficiency evaluation is very limited. I'd like to see latency and throughput evaluation under different context length/batch size workloads.

5. The core design of the work heavily relies on the findings of Induction Heads and Retrieval Heads, making its novelty limited. While I don't think this is much of an issue due to the practical performance gain the proposed method achieved, the relationship of such works deserves much better highlight. The mere mentioning around L167 is not enough.

6. I'd like to see more dataset coverage beyond LongBench — as it is no longer very long w.r.t. models like Llama 3.1. Evaluation of Llama 3.1 on $\infty$Bench would fill this gap.


In summary, I think this is a decent work that cleverly leveraged some observations of retrieval/induction ideas. I am open to increasing my rating should the author provide the requested clarifications and evaluations. I also apologize for the slightly delayed review — I will try to be more engaging during the discussion period.

**Questions:**

DuoAttention and Not All Heads Matter are two concurrent works of RazerAttention. I'd appreciate it if the author could discuss their similarities and differences. In particular, I am interested in Figure 13(1) of DuoAttention — which supposedly features RazorAttention — but the needle performance is very different from your own reporting. I wonder why?

---

> ### Author Response · Authors · 2024-11-24
>
> We sincerely appreciate your constructive feedback of our paper, and we will make certain updates w.r.t. your concerns in our revised version. Below are our responses to your concerns.
>
> Q1: Comparison with SnapKV.
>
> We agree that SnapKV is a competitive approach for KV cache compression, especially when user queries are known before generation. Below, we present the results of SnapKV in both query-aware and query-agnostic settings: • Query-aware: SnapKV demonstrates impressive compression ratios and accuracy under known query scenarios. • Query-agnostic: In scenarios where the user query is not pre-defined, SnapKV’s performance deteriorates significantly, as shown by the following results of Needle in a Haystack (see Section.1 in our supplementary updated):
> | RazorAttention   | SnapKV (query-aware)   |  SnapKV (query-agnostic)   |
> |------------|------------|------------|
> | 98.33 | 100 | 69.75 |
>
> This resilience is due to its training-free design and headwise sparse pattern, which ensure minimal information loss across various query types. We believe this discussion is crucial and will incorporate it into the revised version to emphasize the practical advantages of RazorAttention over query-dependent methods like SnapKV.
>
>
> Q2: Detailed setting of experiments and more long-context benchmarks.
>
> We acknowledge the variation in Needle in a Haystack accuracy (approximately 2%) under different prompts. We will clarify the specific prompt used for RazorAttention and all baselines in our revised version. Meanwhile, Here we include the full results of RazorAttention on Longbench with Model Llama3.1-8B-instruct under the same setting with our paper.  We also report the accuracy of Snapkv at the same compression rate.
>
> | Dataset                  | Baseline | RazorAttention | SnapKV |
> |--------------------------|----------|----------------|--------|
> | 2wikimqa                | 49.25    | 49.81          | 49.12  |
> | hotpotqa                | 57.61    | 57.22          | 57.60  |
> | musique                 | 33.72    | 32.80          | 32.55  |
> | multifieldqa_en         | 55.77    | 56.64          | 56.19  |
> | multifieldqa_zh         | 63.47    | 63.81          | 62.99  |
> | narrativeqa             | 29.23    | 29.54          | 30.03  |
> | qasper                  | 47.53    | 47.32          | 47.61  |
> | triviaqa                | 91.50    | 91.20          | 91.50  |
> | gov_report              | 34.58    | 33.08          | 32.97  |
> | qmsum                   | 25.27    | 25.10          | 25.37  |
> | vcsum                   | 17.28    | 16.99          | 16.49  |
> | dureader                | 34.88    | 31.91          | 31.64  |
> | lcc                     | 24.68    | 24.62          | 24.64  |
> | repobench-p             | 25.57    | 25.36          | 25.33  |
> | passage_retrieval_en    | 99.50    | 100.00         | 99.50  |
> | passage_retrieval_zh    | 90.45    | 95.98          | 90.45  |
> | passage_count           | 10.08    | 9.75           | 9.83   |
> | trec                    | 14.50    | 9.25           | 17.67  |
> | lsht                    | 0.00     | 0.00           | 0.00   |
> | multi_news              | 26.92    | 26.81          | 26.77  |
> | Samsum                  | 13.50    | 13.97          | 13.37  |
>
> As you suggest, we have included the results of infinite-bench and ruler below. Due to time constraints, only a subset of the experiments have been included. We will present additional results upon completion of the remaining experiments.
> | InfiniteBench | Original Model | RA Model |
> |----------------|-----------------|-----------|
> | codedebug       | 21.57          | 21.57     |
> | ensum          | 30.71          | 30.58     |
> | endia          | 19.50          | 19.50     |
> | enqa           | 29.09          | 29.50     |
> | enmc           | 63.31          | 63.31     |
>
>
>   Ruler | Original Model | RA Model
> -- | -- | --
> ruler_niah_single_1_16k | 100 | 100
> ruler_niah_single_2_16k | 100 | 100
> ruler_niah_single_3_16k | 100 | 100
> ruler_niah_multikey_1_16k | 100 | 100
> ruler_niah_multikey_2_16k | 99.11 | 99.11
> ruler_niah_multikey_3_16k | 99.11 | 99.11
> ruler_niah_multivalue_16k | 99.11 | 97.54
> ruler_niah_multiquery_16k | 95.09 | 95.31
> ruler_vt_16k | 80.54 | 86.61
> ruler_fwe_16k | 89.29 | 85.42
> ruler_cwe_16k | 90.09 | 90.27
> ruler_qa_squad_16k | 88.39 | 88.39
> ruler_qa_hotpotqa_16k | 56.25 | 56.25
> ruler_16k | 92.07538462 | 92.15462
> ruler_niah_single_1_32k | 100 | 100
> ruler_niah_single_2_32k | 100 | 100
> ruler_niah_single_3_32k | 100 | 100
> ruler_niah_multikey_1_32k | 100 | 100
> ruler_niah_multikey_2_32k | 96.43 | 96.43
> ruler_niah_multikey_3_32k | 100 | 100
> ruler_niah_multivalue_32k | 97.54 | 94.2
> ruler_niah_multiquery_32k | 88.17 | 91.52
> ruler_vt_32k | 87.86 | 93.04
> ruler_fwe_32k | 88.69 | 88.99
> ruler_cwe_32k | 12.41 | 20.45
> ruler_qa_squad_32k | 86.61 | 85.71
> ruler_qa_hotpotqa_32k | 46.43 | 46.43
> ruler_32k | 84.93384615 | 85.90538

---

> ### Author Response · Authors · 2024-11-24
>
> Q3: The efficiency evaluation.
>
> We evaluate RazorAttention’s decoding latency and throughput on GLM-9B-1M model on 8 Ascend 910B NPUs under different input length for both prefill and decoding:
>
> | Input Length | Batch Size | Prefill Speedup | Decoding Speedup |
> |--------------|------------|-----------------|------------------|
> | 128k         | 1          | 1.2%          | 3.1%           |
> |              | 4          | 5.66%           | 15.9%            |
> |              | 8          | 8.03%           | 26.67%           |
> |              | 10         | 7.05%           | 29.47%           |
> |              | 18         | From OOM to feasible inference |              |
> | 256k         | 1          | 6.58%           | 9.2%           |
> |              | 4          | 8.76%           | 25.47%           |
> |              | 5          | 9.93%           | 29.01%           |
> |              | 9          | From OOM to feasible inference |              |
>
> Below is the maximum throughput improvement of using RazorAttention under different input length:
>
> | Input Length | Maximum Throughput Improvement |
> |--------------|------------|
> |128k | 64.2% |
> |256k | 71.81% |
>
>
> Q4: Comparison with [1] and [2].
>
> We respectfully emphasize that RazorAttention is the first method to utilize a headwise sparse pattern for KV cache compression. Below, we summarize the key differences between RazorAttention and the recent works [1] and [2]:
> 1. Comparison with [2]: In [2], the authors rely on query-based importance metric to select the “important” tokens (Section 3.3 in their paper). While this approach can perform well when the query is known, it introduces the risk of information loss for future queries, just like SnapKV.
> 2. Comparison with [1]: Unlike [1], which employs continual training and knowledge distillation to recover model performance after compression, RazorAttention achieves a nearly lossless compression without any additional training. This training-free design is a more challenging and practical approach. We believe the compensation token we introduced and our head selection method (inductive + echo heads) provide valuable insights into the underlying functionality of LLMs in long-context scenarios.
> 3. Regarding Figure 13(1) in [2]: In [2] authors did not provide specific details about this experiment. We think the discrepancy might be due to their exclusion of echo heads, which we have found to play a critical role in our results. We are confident in the reproducibility of our results.
>
> Q5: Full results of LongBench.
>
> Below we present the full results of Longbench with Llama3.1-8B-instruct:
> | Dataset               | Baseline | RazorAttention |
> |-----------------------|----------|----------------|
> | 2wikimqa             | 49.25    | 49.81          |
> | hotpotqa             | 57.61    | 57.22          |
> | musique              | 33.72    | 32.80          |
> | multifieldqa_en      | 55.77    | 56.64          |
> | multifieldqa_zh      | 63.47    | 63.81          |
> | narrativeqa          | 29.23    | 29.54          |
> | qasper               | 47.53    | 47.32          |
> | triviaqa             | 91.50    | 91.20          |
> | gov_report           | 34.58    | 33.08          |
> | qmsum                | 25.27    | 25.10          |
> | vcsum                | 17.28    | 16.99          |
> | dureader             | 34.88    | 31.91          |
> | lcc                  | 24.68    | 24.62          |
> | repobench-p          | 25.57    | 25.36          |
> | passage_retrieval_en | 99.50    | 100.00         |
> | passage_retrieval_zh | 90.45    | 95.98          |
> | passage_count        | 10.08    | 9.75           |
> | trec                 | 14.50    | 9.25           |
> | lsht                 | 0.00     | 0.00           |
> | multi_news           | 26.92    | 26.81          |
> | Samsum               | 13.50    | 13.97          |
>
> [1]. DuoAttention: Efficient Long-Context LLM Inference with Retrieval and Streaming Heads
>
> [2]. Not All Heads Matter: A Head-Level KV Cache Compression Method with Integrated Retrieval and Reasoning

---

> > ### Author Response · Authors · 2024-11-25
> >
> > Dear reviewer, thank you once again for the time devoted to handling this work. Your insights have been helpful, and we sincerely appreciate your thoughtful review. Please don’t hesitate to let us know if there are any additional questions or points we can further clarify.

---

> ### Comment · Reviewer_jpBx · 2024-12-02
> **Raising my score to 8 but pending a few more requests, and my $0.02 on some other standing issues.**
>
> I appreciate the authors for providing a comprehensive rebuttal. With a proper comparison to SnapKV and broader evaluation coverage, many of my concerns have been alleviated. However, there are still a few areas the authors should better address:
>
> * **Efficiency Evaluation.** The efficiency metrics, gathered on non-standard NPUs, offer very limited insights to the community. It is essential that the authors provide GPU-based efficiency reports, especially for an efficiency-focused paper. My assumption is that the authors currently lack a complete kernel implementation for the proposed method. This is typically a disqualifying factor, as it leaves the end-to-end efficiency of the proposed approach unverified. That said, given that DuoAttention achieves notable efficiency gains under thorough evaluation and has been open-sourced, there is little doubt RazorAttention could achieve similar results, given their nearly identical inference patterns. The authors should consider borrowing the kernel implementation of DuoAttention, crediting it appropriately, and providing comparable efficiency results.
>
> * **Query Dependency vs. SnapKV.** According to the newly added results, there is only a slight performance difference between SnapKV and the proposed method. The authors correctly observe that SnapKV's dropping strategy heavily depends on query awareness (e.g., whether it is within the window). However, the current evaluation does not adequately support this claim. To my knowledge, SharedContextBench (SCB) is the only dataset that evaluates multi-round scenarios in a long-context setting. I recommend that the authors include results from SCB in the final version. SCB has already demonstrated the performance degradation of SnapKV, and given the (seemingly) task-agnostic mechanism of RazorAttention’s retrieval/induction heads, I am confident it would outperform SnapKV.
>
> Normally, I would request these results before adjusting my score. However, in this case, I am inclined to increase my score now due to 1) the limited time available (partially due to my own late response), 2) the confidence that supportive results can be achieved based on findings from concurrent work, 3) SCB’s ongoing review at ICLR, which may delay access to finalized supporting materials, and 4) most importantly, RazorAttention is the first work (and predates others by a considerable margin) to leverage induction/retrieval head findings for KV cache compression. **That said, I ask the authors to confirm that both of these improvements will be included in their camera-ready version or, if SCB’s availability blocks progress, in an arXiv version released soon after.**
>
> ---
>
> Finally, I’d like to share my thoughts on several standing issues raised by other reviewers:
>
> * **Comparison with DuoAttention.** The two methods indeed share many similarities. However, RazorAttention predates Duo/Not All Heads Matter by a significant margin. While the methods are concurrent and likely developed independently, this alone should shield RazorAttention from criticism in this regard.
>
> * **Limited Compression Ratio.** The evaluation already demonstrates that RazorAttention remains effective at a relatively high compression ratio (~3x). While finer-grained compression is a fair request, I see it more as a bonus rather than a core requirement as near-lossless performance at ~3x is already pretty usable. That said, the authors should consider expanding Table 4 to include results with >15% protected heads for completeness.
>
> * **Combining with Other KV Cache Compression Methods.** KV cache compression spans multiple dimensions, and methods addressing each dimension are still evolving. While combining approaches is possible, such combinations are likely out of scope for this paper and, most importantly, probably too heavy of an ask for a paper with seven reviewers already demanding numerous additional results.
>
> ---
>
> One additional question: does DuoAttention *"employ continual training and knowledge distillation to recover model performance **after compression**"*? I was under the impression that Duo does distillation to update its gates, identifying heads worthy of a full cache, but not after the compression. Could the authors please double-check this?

---

> ### Author Response · Authors · 2024-12-03
>
> We sincerely appreciate your valuable feedback. In response, we will incorporate the efficiency results tested on A100 and the performance results on SharedContextBench (SCB) in the updated version of our manuscript. We have conducted the following experiments to address the concerns about the performance of RazorAttention and SnapKV under different compression ratios, and the combination of RazorAttention with other methods.
>
> 1) Performance of RazorAttention and SanpKV under different compression ratios:
>
> Below we present the results for these two methods under 15%, 20%, and 40% KV cache budget for Needle In A Haystack. We use Llama3.1-70b and masked out the queries for both methods (to simulate the situation where the user queries are unknown).
>
> | Needle In A Haystack | RazorAttention | SnapKV |
> |----------------|-----------------|-----------|
> | 15%       | 100%         | 65%    |
> | 20%         | 100%        | 69%    |
> | 40%          |100%         | 78%     |
>
> 2) Combination of RazorAttention and Quantization:
>
> Below we further compress the KV cache under RazorAttention into MX-FP4 on Llama3.1-8B and report its results on LongBench.
>
> | Dataset                  | baseline | KV MXFP4+RA |
> |--------------------------|----------|-------------|
> | 2wikimqa                 | 49.25    | 47.64       |
> | hotpotqa                 | 57.61    | 55.26       |
> | musique                  | 32.72    | 30.7        |
> | multifieldqa_en          | 55.77    | 57.23       |
> | multifieldqa_zh          | 63.47    | 61.2        |
> | narrativeqa              | 29.23    | 30.32       |
> | qasper                   | 47.53    | 47.09       |
> | triviaqa                 | 91.5     | 91.79       |
> | gov_report               | 34.58    | 33.57       |
> | qmsum                    | 25.27    | 25.41       |
> | vcsum                    | 17.28    | 17.29       |
> | dureader                 | 34.88    | 33.62       |
> | lcc                      | 24.63    | 23.74       |
> | repobench-p              | 25.57    | 23.66       |
> | passage_retrieval_en     | 99.5     | 100         |
> | passage_retrieval_zh     | 90.45    | 95.22       |
> | passage_count            | 10.08    | 5.64        |
> | trec                     | 14.5     | 13.04       |
> | lsht                     | 0        | 0.5         |
> | multi_news               | 26.92    | 26.66       |
> | Samsum                   | 13.5     | 13.23       |
>
> 3) How does DuoAttention choose the gate for each head:
>
> In DuoAttention,  authors use continual training together with distillation before compression. This training process determines a score for each head, and they set the heads with a larger score as full-cache attention heads afterward in compression.

---

> > ### Comment · Reviewer_jpBx · 2024-12-03
> >
> > Thanks for the additional results. These seem to primarily address the standing concerns of other reviewers, which I personally do not emphasize much for the reasons listed in my last reply. However, they are still good to have. Regarding those issues, I am more interested in an expanded Table 4.
> >
> > (By the way, you might also want to clarify your needle length for your first table.)
> >
> > Regarding your response on DuoAttention, it appears consistent with my understanding, which makes your earlier *"recover after compression"* comment a bit casual. If you plan to use Duo's kernel for efficiency reporting, please consider providing a more precise and thorough introduction.
> >
> > Overall I think the authors discovered a recipe of great potential, its technical novelty might be a bit slim, but it sure is performant and carry great pratical significance. GL!

---

> > > ### Author Response · Authors · 2024-12-03
> > >
> > > Yes, we will clarify the setting of our efficiency test in detail (probably use OpenAI Triton to implement this hybrid attention together with the compensation token). We sincerely appreciate your thoughtful suggestions, which have allowed us to further enhance our work.

---

### Meta-Review · Area_Chair_5j2K · 2024-12-21

**Metareview:**

The paper introduces RazorAttention, a novel KV cache compression algorithm that selectively retains full information in retrieval heads while condensing remote tokens in non-retrieval heads using a compensation token. This method achieves over 70% reduction in KV cache size without compromising model performance, offering compatibility with FlashAttention and requiring no retraining. The key strengths include its efficient head-specific strategy, lossless compression, and extensive evaluation across diverse language models, showing practical utility and innovation. However, limitations include the reliance on earlier insights about retrieval heads, limited comparisons with some advanced baselines, and a need for more detailed efficiency evaluations on GPUs. Despite these, the paper demonstrates strong practical performance and a significant impact on LLM efficiency, making it a valuable contribution to the field. The decision to accept is based on its substantial improvements in resource efficiency, innovative methods for KV cache compression, and broad applicability.

**Additional Comments On Reviewer Discussion:**

The reviewers initially raised concerns regarding the novelty of leveraging retrieval heads, lack of detailed efficiency evaluations, and limited baseline comparisons. During the rebuttal, the authors clarified their contributions by expanding the definition of retrieval heads to include echo heads and introducing the compensation token. They addressed efficiency by providing results for various hardware setups and compression ratios. Comparisons with advanced baselines like SnapKV highlighted RazorAttention's robustness, particularly in query-agnostic scenarios. The authors committed to addressing remaining gaps in the final version. These efforts successfully resolved major concerns, and the reviewers updated their scores to reflect the paper's merits. The decision weighed the resolution of key concerns and the paper's impact, favoring acceptance.

---

### Decision · Program_Chairs · 2025-01-22

Accept (Poster)